Corrected: Author correction

# A RAB35-p85/PI3K axis controls oscillatory apical protrusions required for efficient chemotactic migration

Salvatore Corallino[1], Chiara Malinverno[1], Beate Neumann[2], Christian Tischer[2], Andrea Palamidessi[1], Emanuela Frittoli[1], Magdalini Panagiotakopoulou[3], Andrea Disanza[1], Gema Malet-Engra[1], Paulina Nastaly[1], Camilla Galli[1], Chiara Luise[4], Giovanni Bertalot[4], Salvatore Pece[4], Pier Paolo Di Fiore (iD) [1,4,5], Nils Gauthier[1], Aldo Ferrari[3], Paolo Maiuri (iD) [1] & Giorgio Scita (iD) [1,5]

How cells move chemotactically remains a major unmet challenge in cell biology. Emerging evidence indicates that for interpreting noisy, shallow gradients of soluble cues a system must behave as an excitable process. Here, through an RNAi-based, high-content screening approach, we identify RAB35 as necessary for the formation of growth factors (GFs)-induced waves of circular dorsal ruffles (CDRs), apically restricted actin-rich migratory protrusions. RAB35 is sufficient to induce recurrent and polarized CDRs that travel as propagating waves, thus behaving as an excitable system that can be biased to control cell steering. Consistently, RAB35 is essential for promoting directed chemotactic migration and chemoinvasion of various cells in response to gradients of motogenic GFs. Molecularly, RAB35 does so by directly regulating the activity of p85/PI3K polarity axis. We propose that RAB35 is a molecular determinant for the control of an excitable, oscillatory system that acts as a steering wheel for GF-mediated chemotaxis and chemoinvasion.

[1] IFOM, the FIRC Institute of Molecular Oncology, Via Adamello 16, 20139 Milan, Italy. [2] Advanced Light Microscopy Facility, European Molecular Biology Laboratory, Meyerhofstr. 1, 69117 Heidelberg, Germany. [3] ETH Zurich, Laboratory of Thermodynamics in Emerging Technologies, Sonneggstrasse 3, 8092 Zurich, Switzerland. [4] Department of Experimental Oncology, European Institute of Oncology, via Adamello 16, Milan 20139, Italy. [5] Department of Oncology and Hemato-Oncology, University of Milan, Via Festa del Perdono 7, 20122 Milan, Italy. Correspondence and requests for materials should be addressed to G.S. (email: Giorgio.scita@ifom.eu)

Cells and particularly tumour cells use different motility modes to disseminate[1]. Each of these modes is driven and controlled by distinct molecular pathways, the nature of which remains largely unexplored. In one such strategy, referred to as mesenchymal motility, single cells may detach from the tumour mass and advance as individual, invasive units[2]. One of the first steps of mesenchymal migration and invasion, particularly in response to growth factors stimulation, is the acquisition of a front-to-back polarity, which is driven by the extension of different kind of actin-based migratory protrusions, including canonical actin-rich flat lamellipodia, small finger-like filopodia[3,4], sausage-like lobopodia[5], blebs[6] and the poorly studied, apically localized circular dorsal ruffles (CDRs)[7]. CDRs have been proposed to be markers of cellular transition from amoeboid to mesenchymal migration[8]. Topologically, CDRs are formed on the dorsal surfaces of the cells. They often initiate in a polarized spot on the membrane, from which they first expand as a ring[7,9], to subsequently contract centripetally, generating a cup-like structure, leading to the formation of macropinosomes[10]. Consistently, these structures are sites of growth factor-induced macropinocytic internalization and promote the endocytosis of various membrane-bound molecules including epidermal growth factor (EGF)[11] and non-ligand engaged β3 integrin[12].

Among the growth factors known to elicit robust and directional migration, hepatocyte growth factor (HGF) in epithelial cells[13] and platelet-derived growth factor (PDGF)[14,15] in fibroblasts, have been shown to be potent and specific inducer of CDRs[7]. Molecularly, the formation of CDRs requires the activation of the respective cognate receptor tyrosine kinases, PDGFR and c-MET, which in turn trigger the recruitment of signalling complexes that invariably lead to the modulation of the actin polymerization[7,9]. A pivotal role, in this context, is exerted by RAC1 (ref. [16]), whose activity must become spatially restricted for CDRs to form[8,17–19]. Additional important factors in the formation of CDRs are lipid kinases, and specifically PI3K as both pharmacologically or genetic inhibition of this enzyme abrogate their formation by preventing the generation of phosphatidylinositol-3,4,5-phosphates important for the recruitment of membrane binding, curvature sensitive Bin-Amphiphysin-Rvs (BAR)-containing proteins[16] as well as to activate RAC1 GEFs, including TIAM1 (refs. [8,17]) and DOCK1 (ref. [20]). Notably, the latter protein has recently been shown to mediate CDRs formation acting specifically downstream of oncogenic forms of KRAS[20]. Activated RAS molecules have, indeed, long been shown to promote CDRs and macropinocytosis[21,22], which is exploited as a mean to scavenge protein and lipid sources in order to refill the amino acid pools, fuel mitochondrial metabolism and lipid biosynthesis[23–26], ultimately fostering survival in nutrient-deprived, tumour microenvironment. Thus, CDRs are emerging as structures that integrate migratory and endocytic processes, and as such can be exploited by certain tumours to enhance their metabolic plasticity and invasiveness. The identification of key molecular determinants governing their formation is, therefore, paramount and likely to have important implications for our understanding of how cells perceive, respond and adapt to soluble environmental cues.

One striking feature of CDRs, in addition to their distinct circular architecture, is their peculiar kinematic behaviour. CDRs are transient, forming only once upon stimulation[7,10], and frequently form in a recurrent wave-like pattern[10]. In addition, they display a polarized distribution, particularly when induced by local gradients of growth factor[10]. These combined kinematic properties suggest that these structures may operate as an oscillating device or steering wheel in driving chemotactic motility. It follows that factors controlling their formation might be critical chemotactic sensor or regulator and promoter of a directional,

mesenchymal mode of motility by specifically controlling growth factor-mediated cell steering.

Here, by exploiting CDRs as an easy-to-follow read out for an automated high-content, imaging-based screening, we identify RAB35 as necessary and sufficient for CDR formation. We further show that RAB35-induced CDRs behave as recurrent and propagating waves. Addition of growth factor induces their polarized formation, suggesting that these structures are indeed actin-based oscillatory systems that might be biased for efficient chemotaxis. Accordingly, RAB35 is involved in efficient chemotaxis in various migratory systems. Molecularly, RAB35 acts downstream of RAB5A/RAC1 pathways by directly binding and controlling the activity of p85α, the regulatory subunit of PI3K. Collectively, our findings uncover a pivotal role of RAB35/p85–PI3K axis in controlling oscillatory CDR protrusions for efficient chemotaxis and macropinocytosis.

## Results

**RAB35 is essential for CDRs formation**. We used CDRs as a read out for an RNAi-based phenotypic screening to identify new critical players promoting their formation and set out to test whether they are acting as oscillating waves steering cells during chemotaxis.

Given the involvement of CDRs in endocytic processes[12,13], we specifically targeted each of the mammalian members of the RAB GTPase family, which includes more than 60 independent genes[27]. RAB GTPases, by controlling key steps of endocytosis and vesicular trafficking, are necessary for the execution of actin-based polarized functions that are in turn essential for cell migration and invasion[27]. We devised a multistep screening strategy. For the primary screening, we used mouse embryonic fibroblasts (MEFs) as model system because in response to PDGF stimulation the large majority of cells form easily detectable and prominent CDRs that appear in a highly synchronous, temporal fashion (with a peak after 10 min of stimulation). We employed a custom siRNA library, targeting all the members of the family (about 60 genes using three individual siRNAs for each gene, Supplementary Data 1), arrayed onto 96-wells imaging plates. We developed a fluorescence-based imaging pipeline to automatically monitor and quantitatively score the formation of CDR in PDGF-stimulated MEFs (for details see Methods and Supplementary Figure 1A–C). The accuracy of our pipeline was evaluated by manually/visually inspecting a subset of randomly chosen images in order to measure the ability to correctly recognize CDRs. The true-positive and false-positive rate of CDR recognition were 0.94 and 0.23, respectively. Raw data were quality controlled by removing images with a low number of nuclei (in all these cases the corresponding siRNA was considered as inconclusive/cytotoxic) and by discarding out-of focus images. In addition, we systematically evaluated plates based on the transfection efficiency of our cells (~80 %) that was measured by counting the percentage of polylobed nuclei upon *Incenp* downregulation (Supplementary Figure 1C). Finally, the efficiency in forming CDRs was normalized in each experimental condition with respect to the negative control (siEGFP) of the screening (Methods) to obtain a CDR score that was used to rank the various treatments (Fig. 1a). The top regulator of CDRs included, as expected, the β isoform of the PDGF receptor, and few RAB GTPases, among which RAB35, the silencing of which resulted in one of the most robust inhibition of CDRs formation (Fig. 1a and Supplementary Data 2). Next, we performed an independent secondary validation step by focusing on those genes for which at least two out of three siRNAs resulted in a CDR score <0.4 (Supplementary Data 2). Gene silencing in these cases was verified by quantitative RTPCR analysis (Fig. 1b). The silencing of

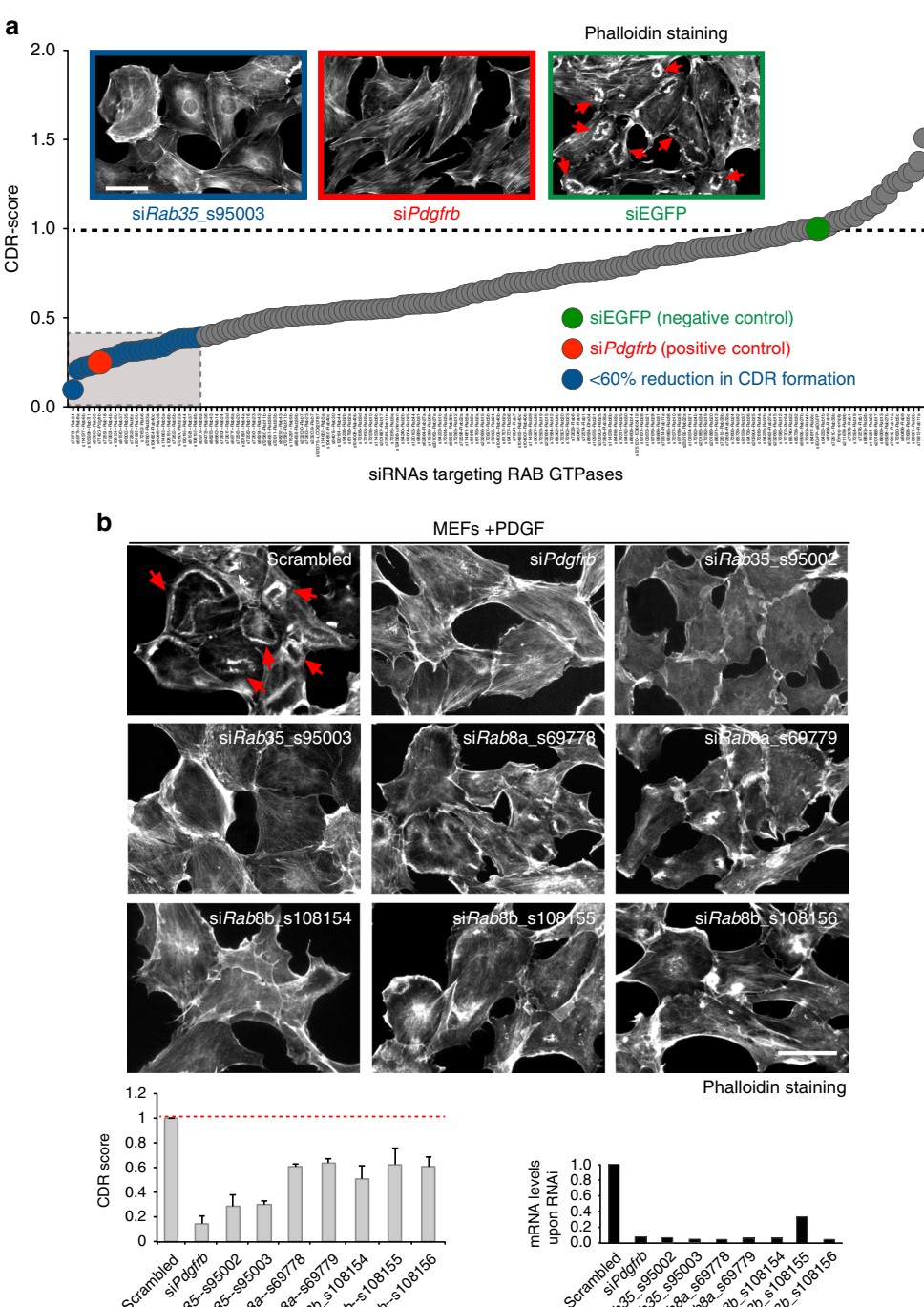

**Fig. 1** siRNA screening for RAB GTPases involved in CDRs formation. **a** siRNAs targeting RAB GTPases were ranked according to their CDR score (Methods and Supplementary Figure 1). Representative images of positive (si*Pdgfrb*) and negative (siEGFP) controls are, respectively, in red and green. siRNAs best inhibiting CDR formation (more than 60% reduction relative to scrambled siEGFP) are enclosed into the light grey box. The oligo s95003 targeting *Rab35*, indicated in blue, is reported as a representative example. Red arrows indicate CDRs. Scale bar, 50 μm. **b** The siRNAs that inhibited CDRs formation more efficiently were validated in a completely independent experiment. Left: MEF cells interfered for *Rab35* (ID: s95002, s95003), or *Rab8a* (ID: s69778, s69779), or *Rab8b* (ID: s108154, s108155, s108156), *Pdgfrb* (positive control), or Scrambled oligo (negative control). Upon PDGF stimulation, cells were fixed and stained with phalloidin. Representative images are shown for each experimental condition. Red arrows indicate CDRs. Scale bar, 50 μm. Right: CDRs were manually counted and normalized against scrambled-transfected, control samples. Data are the mean ± SD (*n* > 200 cells/condition in three independent experiments). The silencing of the targeted genes was verified by qRTPCR

RAB35, RAB8A and RAB8B resulted in a robust and reproducible decreased in CDR activity, thus corroborating the validity of the primary screen and providing evidence that RAB35 is the main regulator of CDR formation among the mammalian RAB protein family (Fig. 1b).

To prove that the CDR-phenotype associated with RAB35 silencing is the result of specific targeting of the gene and rule out off-target effects, we performed a rescue experiment. To this end, we generated a population of MEF cells expressing the HA-tagged human form of RAB35, which is resistant to the

murine oligo used to induce the silencing of the endogenous gene product, in a doxycycline-inducible fashion. Cells interfered for endogenous RAB35 displayed reduced CDR formation, which was fully rescued by the expression of the human protein (Fig. 2a). Of note, ectopically expressed RAB35 was diffuse on the cytoplasm and present on the plasma membrane, but re-localized to CDRs following stimulation with PDGF (Supplementary Figure 1D).

In addition to be migratory, CDRs are also sites of macropinocytic internalization[28]. Real-time phase contrast microscopy, consistently, revealed the formation of large fluid-filled, vesicle-like structures that invariably form following CDR closure (Supplementary Movie 1). This was mirrored by PDGF-mediated increase in the internalization of large molecular weight, fluorescently labelled Dextran, a bona fide macropinocytosis cargo. Importantly, MEFs stably downregulated for RAB35 were

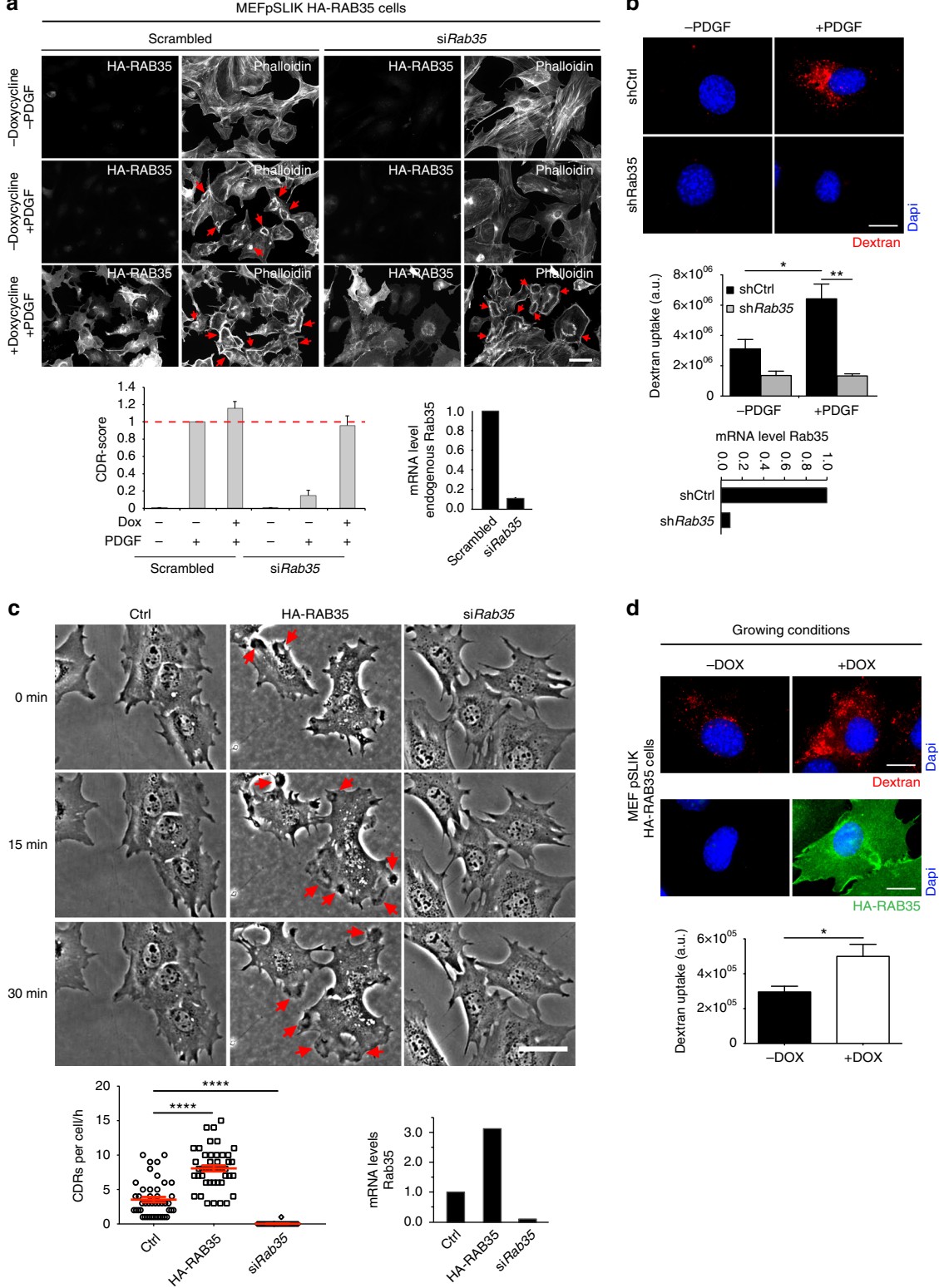

**Fig. 2** RAB35 impairs CDRs formation. **a** An siRNA-resistant human RAB35 rescues CDRs formation. Doxycycline-treated pSLIK-HA-RAB35-human infected MEFs silenced for endogenous RAB35 were serum starved for 2 h, and stimulated with PDGF for 10′. Samples were co-stained with FITC-Phalloidin and the anti-HA antibody to detect F-actin and human HA-RAB35, respectively. Red arrows indicate CDRs. Scale bar, 50 μm. CDR score was calculated by normalizing the number of CDR-positive cells per each condition against the scrambled doxycycline-untreated, PDGF-stimulated sample, used as a control. Data are the mean ± SD ($n > 100$ cells/condition, three independent experiments). The silencing of endogenous RAB35 was verified by qRTPCR. **b** Silencing of RAB35 impairs macropinocytosis. MEF control (shCtrl) and *Rab35*-downregulated (sh*Rab35*) cells were serum starved for 2 h and incubated (+) or not (−) for 1 h with PDGF and tetramethylrhodamine-dextran. Upon fixation, cells were processed for epifluorescence to identify nuclei (blue) and TMR-dextran-positive macropinosomes (red), respectively. Scale bar, 20 μm. Bottom graph, dextran uptake was quantified by determining the total cell fluorescence/cell (see Methods) expressed as A.U. Data are the mean ± SD ($n = 40$ cells/condition, three independent experiments). **$p < 0.01$; *$p < 0.05$. The downregulation of RAB35 was verified by qRTPCR. **c** RAB35 is sufficient to induce the spontaneous formation of CDRs. MEF control (Ctrl), HA-RAB35-expressing (HA-RAB35) and *Rab35*-silenced (si*Rab35*) cells were monitored for 1 h by time-lapse in the absence of PDGF. Still phase contrast images from time-lapse sequence (Supplementary Movie 2) are shown. Red arrows indicate CDRs. Scale bar, 50 μm. The number of CDRs/cell formed in 1 h is expressed as mean ± SEM ($n = 60$ cells/condition, three independent experiments). ****$p < 0.0001$. RAB35 mRNA levels were determined by qRTPCR. **d** RAB35 expression promotes macropinocytosis in the absence of GFs. Doxycycline-treated pSLIK-HA-RAB35-human-infected MEFs were incubated for 1 h with TMR-dextran. Cells were processed for epifluorescence to visualize nuclei (blue), TMR-dextran-positive macropinosomes (red) and the ectopic expression of HA-RAB35 protein (green). Quantification of dextran uptake performed as in **b**. The total fluorescence/cell was expressed as A.U. Scale bar, 20 μm. Data are the mean ± SD ($n = 40$ cells/condition, three independent experiments). *$p < 0.05$. $p$-values are from paired Student's *t*-test

severely impaired in the internalization of dextran (Fig. 2b), reinforcing the notion that this GTPase is critical for CDRs formation and their endocytic functional activity.

We further validated our finding in a different cellular context. We used HeLa cells, which are of epithelial origin and form CDRs in response to HGF stimulation[29]. Silencing of RAB35 in these cells severely decreased CDRs formation as compared to scrambled-transfected cells (Supplementary Figure 1E), indicating that RAB35 is a critical regulator of CDR formation in response to different growth factor-dependent signalling pathways.

Finally, to establish whether RAB35 is also sufficient to induce these structures, we monitored by time-lapse phase contrast microscopy MEF cells expressing HA-RAB35 in a doxycycline-inducible fashion cultured in growing media without any addition of growth factors. Importantly, up-regulation of RAB35 was sufficient to promote the formation of multiple CDRs that display a persistent and rapid dynamics, as confirmed by the quantification of the number of events per cell monitored in 1 h (Fig. 2c and Supplementary Movie 2). This was mirrored by an increased rate of fluorescently labelled dextran internalization (Fig. 2d)

Collectively, these findings identify RAB35 as a non-previously characterized RAB GTPase essential for the formation of CDRs in response to stimulation with various growth factors.

**RAB35-controlled CDRs act as cell steering devices.** CDRs play a key role in cytoskeleton remodelling associated with the transition from sessile to motile states[7]. In addition they frequently, if not invariably, form in close proximity to the cell leading edge[30], and are capable of initiating endo/exocytic cycles of plasma membranes and integrins that are subsequently delivered in a polarized fashion to the prospective lamellipodia[11,12,31,32]. These properties suggest that CDRs may function as bona fide cellular steering compasses to initiate forward and directional chemotactic migration. Given the relationship between CDRs, cell locomotion and cell guidance, we further hypothesize that RAB35 by controlling their formation may also be essential for regulating directional, chemotactic motility.

To act as steering devices CDRs must form in a polarized fashion in response to local gradient of chemotactic growth factors, and their dynamics formation should be spatiotemporally correlated with the extension of lamellipodia protrusions. In addition, structural components and biochemical wiring involved in cell migration guidance are often behaving as excitable oscillatory systems, which may become spatiotemporally biased following chemoattractant exposure[33]. The propagation of actin waves at the ventral surface of neutrophil is a typical case in

point[34–37]. We verified whether CDRs display all these features. Firstly, we monitored by time-lapse microscopy their formation and recorded the subsequent extension of lamellipodia in response to local delivery of PDGF. CDRs formed in polarized directions and their appearance/disappearance was followed by the subsequent extension of flat lamellipodia-like protrusions following the local delivery of PDGF (Fig. 3a and Supplementary Movie 3). The temporal correlation between CDR and leading-edge protrusions was robust in RAB35-expressing cells. RAB35-expressing, but not control cells, form multiple CDRs in the absence of any added growth factor and these structures precede the extension of lamellipodia with a lag phase of about 105″ (Fig. 3b and Supplementary Movie 4). We further exploited the ability of RAB35 to induce a constitutive wave of recurrent multiple CDRs to characterize their overall dynamics and kinematics in more details and relate it to the extension of membrane protrusions. We observed travelling CDR waves, where CDR formed at the rear of an elongated, spatially restricted pseudopodia-like protrusion and move persistently along and in synchrony with the extended protrusion, with nearly identical speed (Fig. 3c and Supplementary Movie 5). We further detected multiple CDRs expanding centrifugally, behaving as iterative waves that form in diverse, but generally peripheral positions expanding centrifugally toward the cell edge (see Supplementary Movies 4). Finally, we detected recurrent and oscillating CDRs waves, which form repeatedly in the same location expanded and enclosed with a typical oscillatory frequency of about 20 min (Fig. 3d and Supplementary Movie 6). Collectively, these features support the notion that CDRs behave as an excitable system of propagating waves, which can be biased by exogenously added PDGF to promote cell steering and chemotactic motility[33].

If CDRs are indeed bona fide, excitable, steering devices, their perturbation should impair directional motility, particularly toward growth factors known to induce robustly their formation right at the onset of chemotaxis. To this end, we first measured the ability of MEFs to migrate through a microporous membrane towards a PDGF gradient in a Boyden chamber assay. Scrambled and RAB35-silenced cells (we used three independent siRNAs) were seeded into Transwell chambers. After 20 h, cells on the top of the membrane were scraped away and the ones migrated at the bottom were stained with crystal violet. An additional time point was taken 3 h after cell plating to demonstrate that we seeded an equal number of cells, which display similar adhesion efficiency across all the different experimental conditions. The loss of RAB35 significantly and robustly reduced the number of cells crossing the porous filter by chemotaxis (Fig. 4a). To monitor MEF chemotaxis in real time and to explore the potential

underlying cellular alterations, we employ a microfluidic commercial device that generate a stable linear gradient of PDGF. We used the ImageJ plugin Chemotaxis tool to extract migratory parameters, including forward migration index (chemotaxis) and mean velocity. Silencing of RAB35 significantly impaired chemotaxis and reduced mean cell velocity (Fig. 4b and Supplementary Movie 7).

To understand whether the altered migratory capability observed upon RAB35 ablation is an intrinsic defect in the molecular machinery sustaining cell locomotion, we also monitored the migratory behaviour of sparsely seeded Ctrl and short hairpin-silenced RAB35 cells in the absence of any external

guiding factor in a random migration assay. Under these conditions, we found that RAB35 loss caused a significant but marginal decrease in cell velocity and had no effect on the formation of lamellipodia protrusions, suggesting that the machinery generating locomotory forces was not strictly dependent on RAB35 (Fig. 4c and Supplementary Movie 8). On the contrary, the stable up-regulation of RAB35, which promoted that extension of multiple and subsequent waves of CDRs, significantly reduced persistent motility (indicated as directionality) and altered mean velocity (Fig. 4c). These latter observations are consistent with the possibility that multiple and short-lived CDRs are induced by the expression of the transgene,

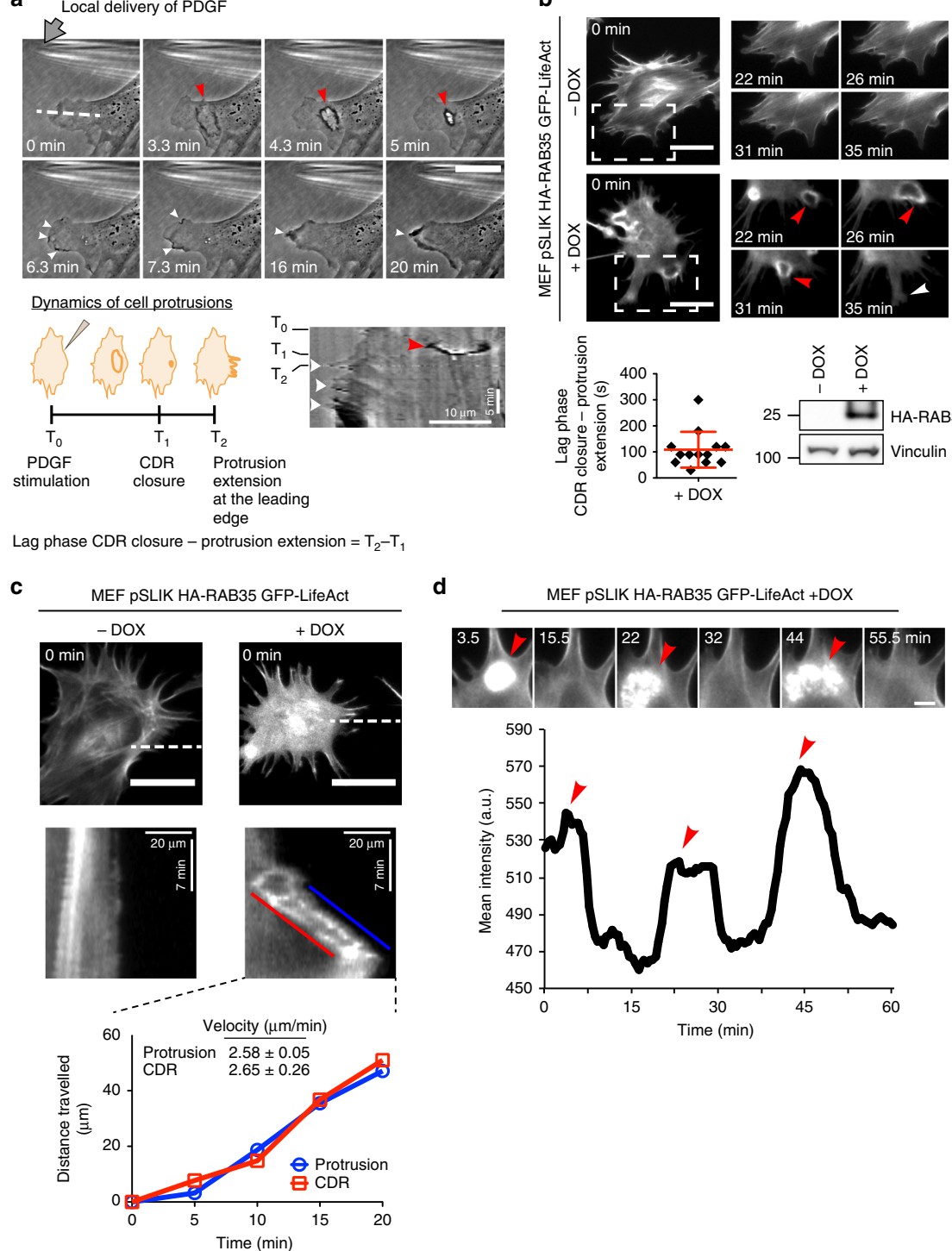

**Fig. 3** Kinematics of CDRs. **a** CDRs formation anticipates protrusion extension with a typical lag phase. Top: Still images from a representative time-lapse (Supplementary Movie 3) of MEFs stimulated by PDGF delivered with a micropipette. Grey arrow indicates the position of released PDGF; red and white arrows indicate CDR and protrusions extension, respectively. The white dashed line indicates the ROI used to perform the kymograph (bottom right panel). Scale bar, 20 μm. Bottom left: Cartoon depicting the dynamics of CDRs and protrusions. Bottom right: kymograph of leading-edge dynamics. $T_0$ PDGF stimulation time, $T_1$ CDR closure time, $T_2$ protrusion extension times. **b** RAB35 promotes the formation of multiple CDRs that expand centrifugally and precede the extension of leading-edge protrusions. Top: pSLIK-HA-RAB35-human-MEFs were infected with EGFP-LifeAct and treated or not with doxycycline to induce transgene expression. Samples were monitored by time-lapse every 30″ for 60′ (Supplementary Movie 4) and still images are shown. Scale bar, 50 μm. Boxes indicate magnified areas shown on the side (Scale bar, 20 μm). Red and white arrows indicate CDRs and protrusion edge, respectively. Bottom left: the lag phase between CDR closure and the subsequent protrusion extension is expressed as mean ± SD ($n = 30$ cells/ experiments, three independent experiments). Bottom right: RAB35 expression was verified by imunoblotting. **c** Travelling CDR waves induced by RAB35. Top: pSLIK-HA-RAB35-human-MEFs were infected with EGFP-LifeAct and treated or not with doxycycline to induce transgene expression. Samples were monitored by time-lapse every 30″ for 20′. Still images from time-lapse sequence (Supplementary Movie 5). Scale bar, 50 μm. The dashed line indicates the ROI used to perform the kymograph. Middle: Kymograph of cell edge (blue) and CDR (red). Bottom: The propagation speed of CDR (red) and leading-edge protrusions (blue) induced by RAB35 is expressed as mean ± SD ($n = 30$ cells, three independent experiments). **d** Oscillating CDR waves induced by RAB35. Doxycycline-treated pSLIK-HA-RAB35-human-EGFP-LifeAct-expressing MEFs were imaged every 30″ for 1 h period. Still fluorescence images from time-lapse sequence (Supplementary Movie 6). Red arrows point the recurrent CDR formation. Scale bar, 10 μm. A representative frequency of recurrent waves. Red arrows indicate peaks that correspond to CDRs

leading to cells that frequently change the direction of their protrusions and motion, and cannot persistently move in a biased direction. To further substantiate this notion, we analysed the migratory behaviour of control and RAB35-silenced MEFs moving along a PDGF gradient through an array of pillars where directional decision choices must be made. The array is composed of pillars from a photocurable hybrid polymer separated by a 4-μm space (Fig. 4d). Control cells navigate toward the PDGF gradient by extending persistent migratory protrusions, most of which were oriented coherently with the direction of the gradient. Conversely, loss of RAB35 reduces significantly chemotaxis and the number of cells with protrusions oriented along the gradient (Fig. 4d).

Finally, if our model were correct, we would expect that RAB35 loss might not be strictly required for migration toward stimuli that poorly or do not induce CDRs formation. Consistently, we showed that RAB35 loss inhibited chemotaxis of MEFs toward serum, which poorly induces CDR formation, much less drastically than toward PDGF (Fig. 5a, b). Additionally, RAB35 was dispensable for chemotactic migration toward EGF, which is unable to induce the formation of CDRs in MEFs (Fig. 5c, d). Similarly, RAB35 loss had no effect on MEFs migration in scratch wound assays. Under these latter conditions, cells migrate by kenotaxis without forming CDRs, while extending lamellipodia and filopodia protrusions like control cells (Fig. 5e and Supplementary Movie 9).

**RAB35 is required for chemoinvasion.** Collectively, our findings argue that deficiency in CDR formation leads to a slight impairment in cell locomotion but a more dramatic inhibition of directional sensing and chemotactic-guided migration further impacting on cell persistence, at least during crawling locomotion on two-dimensional (2D) surfaces typically measured by all these assays. In vivo, however, cell migration occurs within complex three-dimensional (3D) matrices with different structural organization, fibres composition and physical properties. Under these conditions, cells frequently move along single ECM fibres or narrow channels that impose a defined, physical confinement. To this end, we performed migratory assays on one-dimensional (1D) micro-printed as well as suspended ECM lines. These assays mimic 1D tumour interstitial migration and allow, through the live monitoring of cell motion, a precise control of migratory parameters. Using suspended fibres, we showed that RAB35 ablation impeded the crawling mode of locomotion. Cells devoid of RAB35 were no longer able to extend trains of expanding wave and were stuck on the fibres (Supplementary Movie 10). Next, to provide a more reliable quantitation of this phenotype, we turned

to 1D micro-printed fibronectin tracks. We seeded MEF shCtrl and shRAB35 cells on lines of 10 μm in width and monitored their 1D locomotion by time-lapse phase contrast microscopy for 24 h. Cell trajectories were automatically reconstructed by a build in-house ImageJ macro and a number of migratory parameters were extrapolated. Results showed that the absence of RAB35 affected cell velocity, the total length covered and, more relevantly, the persistence of cell motion (Fig. 6a and Supplementary Movie 11).

Next, we wondered whether this protein might also play an active role in tumour cell dissemination in 3D matrix. To address this point, we tested MCF10.DCIS.com cells stably downregulated for RAB35 in a set of in vitro migratory/invasive assays. This cell line is an oncogenic variant of MCF-10A that is widely used to recapitulate the transition from an in situ ductal to an invasive breast carcinoma[38,39]. We first measured the ability of RAB35-depleted cells in migrating through a microporous membrane towards an HGF gradient in a Boyden chamber assay. Ablation of RAB35 profoundly affected the chemotactic migratory ability of MCF10.DCIS.com cells (Fig. 6b). In addition, we also observed that control cells displayed a higher ability to invade and migrate through a thin layer of Matrigel in comparison to RAB35-downregulated cells, as shown in the Matrigel invasion assay (Fig. 6c). To further validate this findings, we monitored by time-lapse phase contrast microscopy the invasive migration of control and RAB35-depleted MCF10.DCIS.com cells into 3D gels of native type I collagen enriched with the motogenic factors HGF[40–42]. Control cells readily invaded the 3D matrix. Conversely the chemoinvasive potential of RAB35-deficient cells was impaired as demonstrated by the reduced invasive forward index and cell velocity (Fig. 6d and Supplementary Movie 12). Finally, we exploited the ability of MCF10.DCIS.com cells to generate invasive outgrowths in 3D basement membrane[43]. Control and RAB35-silenced MCF10.DCIS.com cells were seeded as single cells onto a gel composed of Matrigel and type I collagen and allowed to form spheroids. The addition of HGF in the presence of collagen type I is known to trigger an invasive programme, characterized by the outgrowths of multicellular structures that expand from the regular contour of the spheroids[44]. This transition recapitulates what seen in vivo when in situ ductal carcinoma, confined into the lumen of a duct, convert into invasive ductal carcinoma through the extension of local multicellular protrusions[44]. The percentage of acinar structures that form invasive outgrowths was significantly reduced in RAB35-depleted cells (Fig. 6e), reinforcing the notion that RAB35 is necessary to promote a mesenchymal programme of chemotactic cell invasion in vitro.

**A RAB35–p85–PI3K axis mediates chemotactic responses to PDGF.** What are the cellular processes and molecular pathways RAB35 uses to promote CDR and steer cells in response to chemotactic stimuli?

To address this question, we initially turned to the well-established functional role of RAB35 in controlling clathrin-dependent endocytic internalization and membrane trafficking. More specifically, we tested the possibility that manipulation of

RAB35 levels would impact on the trafficking of growth factor receptors, focusing on PDGFRB. However, no difference in either total PDGFRB or surface amounts could be found following silencing (Supplementary Figure 2A) or ectopic up-regulation (Supplementary Figure 2B) of RAB35.

We next utilized a molecular epistasis approach to position RAB35 action on known pathways controlling CDRs[7]. We have previously shown that the formation of these structures requires

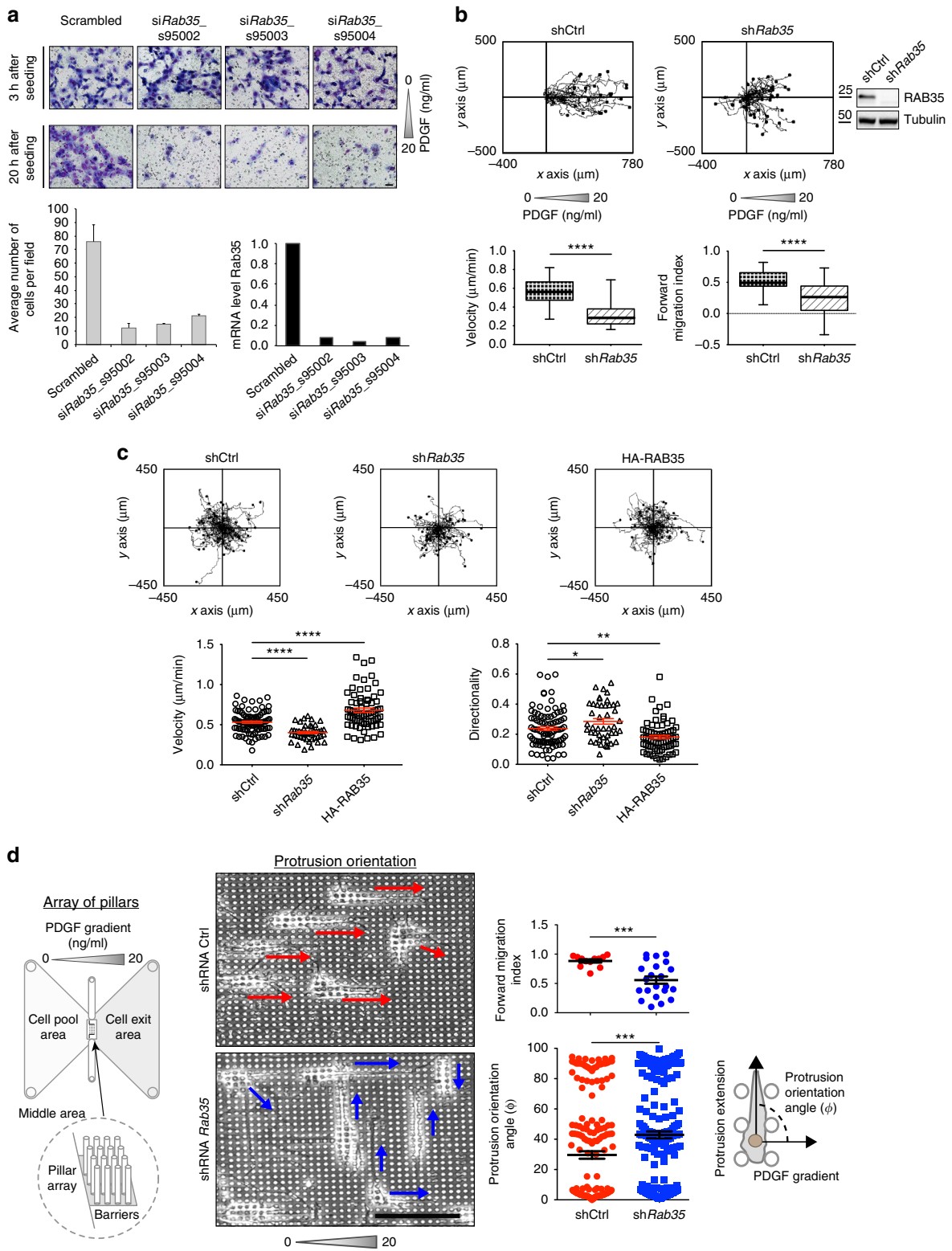

**Fig. 4** RAB35 is required for optimal directional motility and chemotaxis. **a** Transwell assays migration toward PDGF (20 ng/ml) of scrambled and *Rab35*-silenced MEFs. Crystal violet staining was done after 3 h to detect total number of seeded cells, and after 20 h to detect migrated cells to the lower side. The number of cells/field on the lower side is the mean ± SD (>6 fields of view/condition). Gene silencing was verified by qRTPCR. Scale bar, 50 μm. **b** Time-lapse of control (shCtrl) and *Rab35*-silenced (sh*Rab35*) MEFs moving toward PDGF (Supplementary Movie 7). Manually tracked cells were analysed by Chemotaxis Tool ImageJ software plugin. Velocity and forward migration index are shown as whiskers box plots. Solid horizontal line = median value; box outlines = 25th–75th percentiles; whiskers = min and max values. ****$p < 0.0001$. Silencing was verified by immunoblotting. **c** Time-lapse analysis of sparsely seeded, control (shCtrl), *Rab35*-silenced (sh*Rab35*) and HA-RAB35-expressing (HA-RAB35) MEFs (Supplementary Movie 8). Cells were manually tracked to calculate velocity and directionality (Euclidean distance/travelled distance). Mean ± SEM is reported in red ($n > 40$ cells/condition, three independent experiments). ****$p < 0.0001$; **$p < 0.01$; *$p < 0.05$. **d** Chemotactic migration through an array of pillars of control (shCtrl) and *Rab35*-downregulated (sh*Rab35*) MEFs. Left: scheme of the chemotactic device. An 4 μm inter-spaced micropillar array was inserted into a sticky-Slide Chemotaxis chamber (Ibidi). Cells seeded in the cell pool area migrated over a PDGF chemotactic gradient. Cells motility was monitored by time-lapse. Middle: still images of the time-lapse sequence. Superimposed red (shCtrl) and blue (sh*Rab35*) arrows indicate cell protrusion orientation. Scale bar, 50 μm. Top right: manually tracked cells were analysed by Chemotaxis Tool ImageJ software plugin. Forward migration index is reported as mean ± SEM. Bottom right: the protrusion orientation angle (φ) of migrating cells, delimited by the protrusion extension and the direction of the PDGF gradient, was calculated at different time point and is the mean ± SEM. Values of 0 and 90 indicate that protrusions are oriented along and parallel to the chemotactic gradient, respectively ($n > 100$ cells/condition, three independent experiments). ***$p < 0.001$. *p*-values are from paired Student's *t*-test

an active PDGFR, which function as a first-line sensor and a transducer of PDGF signalling, and a functional endosomal trafficking route, in turn, necessary to spatially restrict RAC1-activity and RAC1-dependent actin remodelling into CDR[8], defining an epistatic relationship between these components (Fig. 7a). To position RAB35 along this pathway, we exploited the finding that elevation of the levels of this protein is sufficient to promote multiple CDRs in the absence of GF. Specifically, we monitored by time-lapse phase contrast microscopy the dynamics of CDRs in MEF cells silenced for *Pdgfrb*, or *Rab5a,* b and c or *Rac1* in control (−DOX) and doxycycline-inducible, RAB35-expressing populations (+DOX). The number and dynamics of CDR formation, as expected, was robustly increased upon elevation of RAB35 levels. The silencing of PDGF receptor had no effects on CDR, consistently with the notion that elevation of RAB35 is sufficient to bypass the need of ectopic addition of growth factor to induce these protrusions. On the contrary, silencing of *Rab5* or *Rac1* genes nearly completely abrogated RAB35-induced CDRs (Fig. 7b and Supplementary Movie 13). Thus, RAB35 appears to act downstream of PGDF receptor and either upstream or in a parallel RAB5/RAC1 pathway.

To investigate further this latter possibility and gain a molecular understanding of the mechanisms of action of RAB35, we systematically silenced all, the so far-identified, molecular effectors (Ocrl, Rusc2, Acap2, Mical1, Micall1, Fscn1 and p85) as well as guanine nucleotide exchange factors (Dennd1a, Dennd1b, Dennd6b and Flc) of this GTPase[45,46]. MEF cells were systematically downregulated for the indicated genes [only in the case of p85 we used MEFs double KO of the two isoforms p85[47] instead of performing the transient down-regulation of the genes], serum starved for 2 h, stimulated with PDGF and scored for the ability to form CDRs. The resulting CDR score showed that the genetic ablation of p85 isoforms was the sole manipulation able to phenocopy the loss of RAB35 (Fig. 7c). The lack of effects of critical downstream effectors mediating RAB35 known role as modulator of endocytosis further strengthened the notion that this function may not be the one used to control CDR formation.

To provide evidence of a direct causal link between p85/PI3K axis and RAB35, we performed three sets of experiments. Firstly, we inhibited PI3K activity, which strictly depends on its association with the regulatory subunit p85α and p85β, using a pharmacological inhibitor. Treatment of PDGF-stimulated with LY294002 or the more specific GDC-0941 PI3K inhibitors, the efficacy of which was tested by measuring the phosphorylation levels of the PI3K target and effector protein AKT, severely abrogated the formation of CDRs (Supplementary Figure 3A–B). Inhibition of AKT with MK-2206 had, instead, no effect on PDGF-induced CDRs formation nor on macropinocytosis

(Supplementary Figure 3C). More importantly, pharmacological inhibition of PI3K also effectively abrogated CDRs induced by the sole expression of RAB35 (i.e. in the absence of PDGF addition) (Fig. 8a and Supplementary Movie 14). We corroborated the latter findings using MEF-KO for p85α and β. These cells and the related control cells were engineered to express RAB35 in a doxycycline-inducible fashion and monitored by time-lapse phase contrast microscopy. Removal of p85α and β completely prevented the formation of highly dynamic CDRs induced by RAB35 expression (Fig. 8b and Supplementary Movie 15). Finally, we tested biochemically whether RAB35 acts by directly modulating p85/PI3K activity. Indeed, while we were working on this project, RAB35 was identified as a critical and direct activator of the p85/PI3K–AKT pathway, shown to directly interact with the regulatory p85α subunit and to mediate, through this pathway, cell transformation[48]. In agreement with this latter finding, we found that silencing of RAB35 reduced significantly the phosphorylation of AKT in response to PDGF stimulation, without affecting PDGFR phosphorylation status, and PDGFR-dependent MAPK activity (Fig. 8c). We also showed that the ectopic expression of RAB35, in doxycycline-stimulated MEF pSLIK-HA-RAB35 cells cultured in growing conditions without addition of growth factors, caused the hyper-activation of AKT signalling without affecting the phosphorylation levels of other transducers (Fig. 8d). Finally, RAB35 and p85α co-immunoprecipitated. This interaction was enhanced upon growth factors stimulation. Importantly, wild-type RAB35, but not an inactive dominant-negative RAB35S22N mutant, associated with endogenous p85α, whereas dominant active RAB35 interacted with p85α in a constitutive growth factors-independent fashion (Fig. 8e, f). These latter findings imply that stimulation with growth factors might increase RAB35-GTP levels, leading to the activation of p85/PI3K pathway. Consistent, with this notion a dominant-negative RAB35S22N mutant abrogated PDGF-induced CDRs formation and directional migration (Supplementary Figure 3D), whereas two recently identified activated, tumour-associated RAB35 mutants, RAB35A151T and F161L[48], promoted CDRs formation and elevated AKT phosphorylation in the absence of growth factor stimulation (Supplementary Figure 3E and Movie 16). In addition, there is a correlation between the levels of RAB35 and of phosphoAKT in prostate cancer cell lines (Supplementary Figure 4A–B). Of note, the analysis of the TCGA data set indicated that RAB35 display a significant elevated copy number variation in about 16% of human prostate cancers (Supplementary Figure 4C). The variable expression of RAB35 was also observed across a panel of human prostate cancer in a Tissue Microarray (TMA) with 12/56 (21.4%) adenocarcinoma displaying elevated levels of RAB35 (score ≥ 1.5), whereas only 2/32 (6%) of normal prostate tissues displayed

RAB35 scores = 1.5 (Supplementary Figure 4D). These latter results indicate that deregulation of the levels of this GTPase may be positively selected in a subset of prostate tumours.

## Discussion

How cells respond to chemotactic cues and more specifically how they interpret relative shallow gradients frequently embedded in a highly noisy environment is an issue that has long been fascinated

cell biologists. One emerging law in chemotaxis is that cells to precisely guide their motion must make use of excitable, self-oscillating systems[33,49]. Actin-based migratory protrusions that randomly oscillate extending radially in crawling locomotory cells[50–52] or centrifugally expanding, ventrally restricted actin-rich waves in neutrophil abide to this basic rule[34,53]. These oscillating systems can, in addition, be biased to form in recurrent and polarized fashion in response to chemotactic cues[33,49].

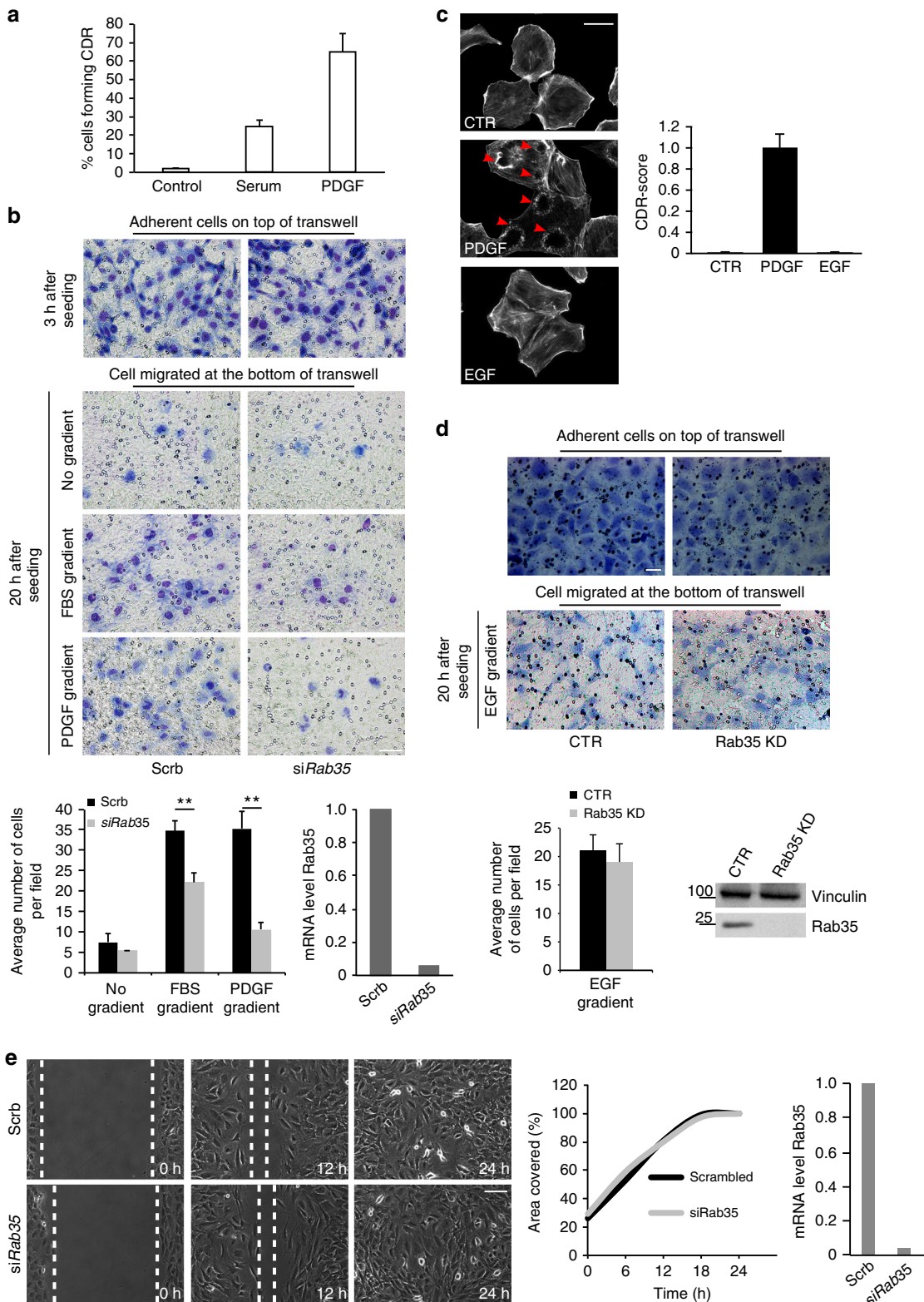

**Fig. 5** RAB35 is dispensable for chemotaxis when CDR do not form. **a** Serum-starved MEF were stimulated with PDGF (20 ng/ml) or 10% serum for 10′, fixed and stained with FITC-Phalloidin to detect F-actin and CDRs. We plotted % cell forming CDRs. Data are the mean ± SD (n = 120 cells/experiments, three independent ones). **b** Scrambled (Scrb) and *Rab35*-silenced (siRab35) MEFs seeded into 8-μm pore Transwell with PDGF (20 ng/ml) or 10% serum added in the lower chamber. Crystal violet staining was done after 3 h to detect total number of seeded cells, and after 20 h to detect migrated cells to the lower side. Chemotaxis was quantified as number of cells/field ± SD (>6 fields of view/condition, in two independent experiments). Data are the mean ± SD. **p < 0.01. Gene silencing was verified by qRTPCR. Scale bar, 50 μm. **c** Serum-starved MEFs were stimulated with PDGF (20 ng/ml) or EGF (100 ng/ml) for 10′, fixed and stained with FITC-Phalloidin to detect F-actin and CDRs. Red arrows indicate CDRs. CDR score was calculated by normalizing the number of CDR-positive cells per each condition against the PDGF-stimulated sample, used as control. Data are the mean ± SD (n > 100 cells/condition in two independent experiments). Scale Bar, 20 μm. **d** Scrambled (CTR) and *Rab35*-silenced (Rab35 KD) MEFs were seeded into 8-μm pore Transwell. PDGF (20 ng/ml) or EGF (100 ng/ml) was added to the lower chamber. Cells migrated through the transwell were stained counted as described in **b**. Scale bar, 40 μm. The extent of chemotaxis was quantified as the average number of cells/field ± SD (>10 fields of view/condition, in two independent experiments). Data are the mean ± SD, **p < 0.01. RAB35 downregulation was assessed by immunoblotting. Vinculin is a loading control. **e** Confluent scrambled (Scrb) and *Rab35*-silenced (siRab35) MEFs were scratched with a pipette tip and recorded by time-lapse. Still images at the indicated time point are shown (Supplementary Movie 9). The percentage of free area covered was determined with ImageJ tool. A representative experiment out of four is shown. Gene silencing was verified by qRTPCR. Scale Bar, 100 μm

CDRs have also been proposed to displays similar features[7,31,54]. Indeed, mathematical and biophysical models support the notion that the growth and decay of CDR actin can be explained as pulse propagation in an excitable media, in which a wave is able to propagate in a nonlinear dynamical system, which is the excitable media[7,54]. However, whether CDRs are linked to chemotactic directional motility and the molecular determinant driving their kinematics has remained, by and large, elusive. Here, we provide evidence that the small RAB35 GTPase is necessary and sufficient to control the formation of CDRs and promote their oscillating, recurrent dynamic behaviours (Fig. 9). We showed that the elevation of RAB35 is sufficient to induce the formation of multiple CDRs that expand centrifugally, travel along elongated protrusions, frequently in the form of oscillating waves, that precede or accompany the extension of lamellipodia protrusions. Stimulation with growth factors, including PDGF in fibroblast and HGF in epithelial cells, biased this behaviour, virtually hijacking this excitable system for efficient chemotactic motility. Not surprisingly, RAB35 is essential for chemotaxis in 2D, directional locomotion in 1D and chemoinvasion in 3D in various cellular systems.

Intriguingly, at the molecular levels, RAB35 controls the activity of an extensively studied chemotactic p85/PI3K axis[55] (Fig. 9). The absolute requirement of this axis in chemoguidance remains somewhat controversial. Nevertheless, p85/PI3K signalling node has been involved both in initiating as well as in biasing and stabilizing branched protrusions, which have been proposed to form stochastically, depending on cellular context[55]. In the case of CDRs formation, the functional requirement of PI3K has long been established[56,57], and analysis of its activity and of coupled lipid-phosphatases has revealed how this axis generate a complex spatiotemporal dynamics of phosphatidylinositol-3,4,5-trisphosphate and its derivate products phosphatidylinositol 4,5-trisphosphate precisely along the membrane confining CDRs[57,58]. This finding supports the notion that intrinsic PI3K-based feedback mechanisms control the duration, extent and the spatial choreography of phospholipids, in turn necessary to modulate the kinematics of CDRs. Within this context, RAB35, which naturally cycles between inactive and active GTP, appears to be perfectly suitable to act as a locally excitable GTPase that directly impinges onto PI3K signalling network, ultimately tuning its activity in a growth factors-dependent fashion. A corollary of this tenet is that RAB35 should be locally accumulating in CDRs, as we showed, and its activity might be spatially restricted at these sites. Whether this is the case, and the additional factors modulating RAB35 activity remains to be established. Nevertheless, collectively, our findings reveal that RAB35 is a key molecular component of an oscillating, excitable network required to set the steering compass of chemotactically migrating cells (Fig. 9).

RAB35 has recently been shown to be a potentially oncogenic RAB protein[48]. This function was shown to be mediated by the ability of RAB35 to regulate a p85/PI3K–AKT axis, which, in turn, impinges onto PDK1 and mTORC2 pathways[48]. Consistently, two somatic RAB35 mutations found in human tumours generate alleles that constitutively activate PI3K/AKT signalling, suppress apoptosis and transform cells in a PI3K-dependent manner[48]. Our finding indicates that RAB35 and the oncogenic-associated mutant forms are also implicated in CDR formation and chemotactic migration in a AKT-independent, but PI3K-dependent fashion. These results argue that certain tumours might specifically exploit and select for alterations of RAB35 also to increase their invasiveness and ability to navigate through complex interstitial environment. We further showed that RAB35 elevation is sufficient to enhance macropinocytosis. This process has recently emerged as a major scavenging route for proteinaceous material and lipid sources in order to refill the amino acid pools, fuel mitochondrial metabolism and lipid biosynthesis, particularly in tumours bearing KRAS or PI3K activating mutations[23,59], ultimately enabling survival in a nutrient-deprived tumour microenvironment. Thus, RAB35 might not only be important for the onset of tumorigenesis, but also to increase nutritional tumour adaptation and progression, possibly by promoting tumour dissemination potential.

## Methods

**Plasmids antibodies and reagents.** Mouse RAB GTPases siRNA library together with experimental controls used in the screening were provided by Ambion (see Supplementary Data 1 for details). Doxycycline-inducible pSLIK-hygromycin (hygro) lentiviral vector carrying the human RAB35 sequence fused to the HA-tag was obtained by Gataway Technology (Invitrogen), following the manufacturer's protocol. Lentiviral expression construct pRRL-Lifeact-EGFP-puromicin (puro) was a gift of Olivier Pertz (University of Basel, Basel, Switzerland). Lentiviral expression constructs Mission shRNA TRC2-pLKO-puro for stable down-regulation of RAB35 (SHCLNG-NM_198163, TRCN0000100534, Mouse; SHCLNG-NM_006861, TRCN0000380003, Human) and pLKO.1-puro Non-Mammalian shRNA Control (SHC002) were purchased from Sigma-Aldrich. GFP-RAB35 WT, -RAB35 S22N and -RAB35 Q67L expression constructs were kind gifts from Cécile Gauthier-Rouviere. siRNA oligos targeting human RAB35 (siRNA IDs: s21707; s21708; s21709) and RAB35 GEFs and Effectors were purchased from Ambion (siRNA IDs: s105812; s116366; s87812; s103715; s115827; s97565; s95591; s101197; s233383; s65847). siGENOME SMARTpool targeting mouse RAC1 (19353) was from Dharmacon.

Mouse monoclonal antibodies anti-phospho-ERK$_{1-2}$ (Thr202/Tyr204, #9106 dilution 1:200), anti-ERK$_{1-2}$ (#9102, dilution 1:500), rabbit monoclonal antibodies anti-phospho-AKT (Ser473, #4058, dilution 1:1000), anti-phospho-PDGFRB (Tyr1009, #3124, dilution 1:50), anti-PDGFRB (#4564, dilution 1:500) and the rabbit polyclonal antibody anti-AKT (#9272, dilution 1:500) were purchased from Cell Signaling. The rabbit polyclonal antibody anti-RAB35 (#113292-AP, dilution 1:600) was from ProteinTech. The rabbit polyclonal antibody anti-p85α (#sc-423, dilution 1:500) was from Santa Cruz Biotechnology and the mouse monoclonal antibody anti-p85 (#05-212, dilution 1:1000) was from Merck Millipore. The anti-Mouse CD140b (PDGF Receptor b, dilution 1:200) PE for FACS analysis was from eBioscience (#12-1402-81). Mouse monoclonal antibodies raised against α-tubulin

(#T5168) or Vinculin (#V9131) and the rabbit polyclonal anti-GFP antibody (#SAB4301138) were from Sigma-Aldrich and they were all used at 1:500 dilution. Mouse monoclonal anti-HA was homemade and derived from HA hybridoma HA-1951. Secondary antibodies conjugated to horseradish peroxidase were from Cell Signaling (#7074, #7076, dilution 1:5000). Cy3-secondary antibodies from Jackson ImmunoResearch (#711-165-152, #715-165-150, dilution 1:400), Dapi (#D-1306) and AlexaFluor 488 (#A-11055, #A-21202) were from Thermo Fisher Scientific. FITC-conjugated Phalloidin (#P5282), LY294002 (#L9908), Doxycycline Hyclate (#D9891) and Fibronectin (#F1056) were from Sigma-Aldrich. The MK-2206 (#11593) and GDC-0941 (#11600) were from Cayman Chemical. Dextran, Tetramethylrhodamine, 70,000 MW (#D1818) used in the macropinocytosis assay

and CellMask Deep Red Plasma membrane Stain (#C10046) were purchased from Thermo Fisher Scientific. Human Recombinant PDGF-BB (#GRF-10694) was from Immunological Sciences; Human Recombinant HGF (#CYT-244) from Prospec; Human Recombinant EGF (#BPS-90201-3) from Vinci-Biochem; Human Insulin (#11376497001) from Roche; Hydrocortisone (#H0888-1G) from Sigma-Aldrich. Lipofectamine RNAiMAX (#13778030) and Lipofectamin 2000 (#11668019) were from Thermo Fisher Scientific.

**Cell cultures and stable cell lines generation.** MEFs were generated from wild-type C57BL/6J embryos at mid gestation and immortalized through passaging[60];

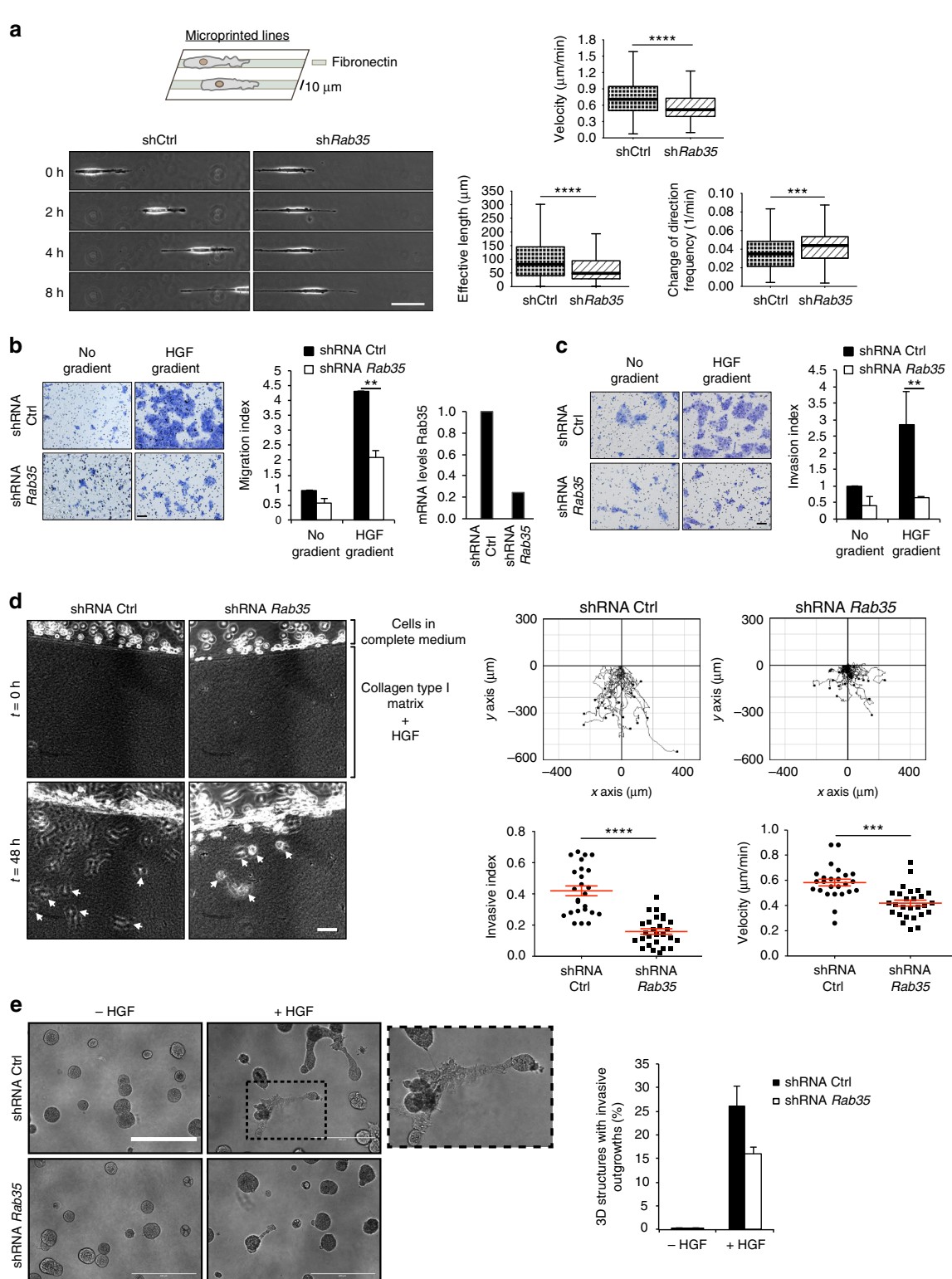

**Fig. 6** RAB35 is required for 1D migration and chemoinvasion in 3D. **a** Still images of control (shCtrl) and *Rab35*-silenced (sh*Rab35*) MEFs migrating on 10-μm wide fibronectin-coated lines (Supplementary Movie 11). Scale bar, 100 μm. Velocity, effective length and change of direction frequency are automatically calculated and shown as whiskers box plots. Solid horizontal line = median value; box outlines = 25th–75th percentiles of SD; whiskers = min and max values. ****$p < 0.0001$; ***$p < 0.001$. **b** Migration toward HGF of control (shRNA Ctrl) and *Rab35*-silenced (shRNA *Rab35*) MCF10.DCIS.com cells in transwell assays as shown in Fig. 4a. The migration index was calculated by normalizing the number of cells/field with respect to control (without gradient). Data are the mean ± SD (>6 fields of view/condition). Gene silencing was verified by qRTPCR. **$p < 0.01$. **c** The chemoinvasive ability toward HGF (100 ng/ml) of control and *Rab35*-depleted MCF10.DCIS.com cells assessed in Matrigel-coated Transwell invasion assays. Cells migrated through the Matrigel layer were stained with crystal violet and invasion index was calculated as in **b**. Data are mean ± SD (>6 fields of view/condition, in three independent experiments). **$p < 0.01$. **d** Control and *Rab35*-silenced MCF10.DCIS.com cells were placed on one side of a chamber slide in which 2.3 mg/ml of acid extracted-only type I collagen gel containing 100 ng/ml HGF and 100 ng/ml EGF was polymerized. Left: still phase contrast images from time-lapse sequence (Supplementary Movie 12). Invading cells are indicated by white arrows. Scale bar, 100 μm. Right: Migration tracks of invading cells over 48 h. Quantification of cell invasion is the mean forward invasion index ± SEM and mean velocity ± SEM (>30 cells/experimental condition, three independent invasion assays). ****$p < 0.0001$; ***$p < 0.001$. **e** Control and *Rab35*-silenced MCF10.DCIS.com cells were grown on a thick Matrigel/type I collagen mixture and overlaid with 2% Matrigel-containing media. After the formation of 3D spheroids, cells were cultured in the presence (+) or absence (−) of HGF for 48 h. Magnification of the boxed area shown on the right. The percentage of 3D structures with invasive outgrowths was quantified and expressed as mean ± SD ($n = 60$ spheroids/condition, two independent experiments). Scale bar, 400 μm. *p*-values are from paired Student's *t*-test

HeLa and DU145 prostate cancer cells were from ATCC. Cells were maintained in DMEM high glucose medium (Lonza) supplemented with foetal bovine serum (10%) and 2 mM L-glutamine. Double p85 alpha and beta KO MEFs were from Dr. Saskia Brachmann. MEFs were derived from embryos of p85α−/− and p85β−/− and wild-type mice (129XC57BL/6). The generation of p85α−/− double KO mice and the establishment of the embryonic fibroblasts from crosses with p85α−/+, p85β−/− mice has been performed previously and described in details in ref. [61]. In brief, the embryos were transferred onto gelatinized tissue culture dishes and the MEFs were immortalized after few passages using SV40 large T antigene expressed as a retroviral vector[47]. The genotypes of the cells were determined by PCR. MEFs p85α−/−p85β−/− were reconstituted by expressing a p85α construct[61]. MCF10. DCIS.com cells were kindly provided by Dr. John F. Marshall (Barts Cancer Institute, Queen Mary University of London) and maintained in DMEM/F12 medium (Invitrogen) supplemented with 5% horse serum, 0.5 mg/ml hydro-cortisone, 10 μg/ml insulin and 20 ng/ml EGF. PC3 prostate cancer cells were cultivated in Ham's F12 with foetal bovine serum (10%).

All cell lines have been authenticated by cell fingerprinting and tested for mycoplasma contamination. Cells were grown at 37 °C in humidified atmosphere with 5% $CO_2$. MEF cells were infected with pSLIK-hygro-HA-RAB35, pRRL-Lifeact-EGFP-puromycin and TRC2-pLKO-puro shRAB35 lentiviruses and selected with the appropriate antibiotic (Hygromycin-B 300 μg/ml and Puromycin 2 μg/ml) to obtain stable and inducible cell lines. MCF10.DCIS.com cells were infected with TRC2-pLKO-puro shRAB35 and shCtrl lentiviruses and selected with Puromycin 2 μg/ml to obtain control and RAB35 stably silenced cells. Packaging of lentiviruses was performed following standard protocols. Four batches of viral supernatant were collected and filtered through 0.45 μm filters. Cells were subjected to four cycles of infection. RAB35 ectopic expression in MEF pSLIK-HA-RAB35 cells was induced by Doxycycline Hyclate (2.5 μg/ml).

**Production of ready-to-transfect 96-wells plates**. The siRNA transfection solution was prepared as in ref. [62]. Briefly, into each well of a 96-wells plate, 5 μl of siRNA stock solutions, with a concentration of 3 μM, were added to 7 μl of transfection reagent solution (3 μl of OptiMEM containing 0.4 M sucrose + 1.75 μl of $H_2O$ + 1.75 μl Lipofectamine 2000) and mixed thoroughly. After 20 min of incubation at RT, 7 μl of 0.2% gelatin solution were added to each well and the resulting transfection mixes diluted 1:50 with water (1 + 49 μl, respectively). The siRNA transfection solution (50 μl) was distributed into empty glass-bottom 96-wells imaging plates (Greiner, Item-No. 655891). In total, 10 replicates for each plate layout were produced and immediately dried in a multi-wells Speed Vac for 2.5 h at medium drying force. "Ready to transfect" 96-wells plates were stored in plastic boxes containing drying pearls orange (Fluka) until usage.

**Primary screening for identifying CDR-related RABs**. MEF cells were seeded into the "ready to transfect" 96-wells imaging plates (800 cells/well) using the cell seeding device Multidrop Combi (Thermo Scientific). After 48 h of incubation at 37 °C and 5% $CO_2$, cells were serum starved for 2 h, stimulated with PDGF (20 ng/ml) for 10 min and fixed in 4% paraformaldehyde (PFA). Fixed samples were permeabilized with 1× phosphate-buffered saline (PBS) + 0.1% Triton X-100 + 1% bovine serum albumin (BSA) for 20 min, followed by FITC-Phalloidin staining (3 μg/ml, for 1 h), Dapi staining (0.5 ng/ml, for 10 min) and CellMask Deep Red staining (1:5000, for 10 min).

Images were acquired with an automated epifluorescence microscope (IX-81; Olympus-Europe) equipped with a stabilized light source (MT20, Olympus-Biosystems, Munich), a firewire camera (DB-H1, 1300 × 1024, Olympus-Biosystems), an objective (Plan10x, NA 0.4; Olympus-Europe) and filter sets from ChromaInc. An image-based autofocusing routine was used to focus on the maximum number of interphase cells (scoring size, intensity, contrast) in a field of view (Dapi channel). Images of nine sub-positions per well were acquired at three

wave-lengths in order to image Phalloidin, CellMask and Dapi stainings with exposure time of 500, 50 and 5 ms respectively.

Typically, upon 48 h of interference, MEFs were serum starved for 2 h followed by stimulation with PDGF. Samples were subsequently processed for immunofluorescence analysis using FITC-Phalloidin to visualize the actin-rich structures of interest, CellMask Deep Red staining, a membrane dye suitable for automated cell segmentation, and Dapi to obtain information about nuclear morphology and cell density. Images were automatically acquired and analysed by a dedicated-image analysis pipeline to quantitatively assess CDRs formation across the different experimental conditions and generate a list of putative candidate genes regulating these structures (Fig. S1A). To evaluate the reliability of the primary screening, experimental controls characterized during the assay optimization phase were distributed on the pre-coated 96-well plates. More specifically, we included: siRNAs targeting *Plk1* and *Incenp*, two genes involved in mitotic segregation, the knock down of which results in obvious morphologically altered nuclei, as transfection efficiency controls; scrambled sequences targeting no endogenous genes in the mouse genome and thus not affecting CDR formation (siEGFP), and *Pdgfrb* silencing oligos, which abrogate CDRs, as negative and positive controls, respectively. We carried out at least three biological replicates for each plate layout acquiring nine field of view for each well and three different fluorescent channels (GFP, Cy5 and Dapi) per position.

**Image and data analysis of the primary screen**. The image and data analysis was performed using the open-source software packages ilastik[63], CellProfiler[64] and R (https://www.r-project.org/). To analyse Phalloidin images and distinguish between pixels that belong to CDRs from other actin-rich structures such as stress fibres, we trained a pixel-based classifier in the supervised pixel classification software ilastik. We manually labelled pixels in 23 representative images from our screen, covering different treatments and intensity levels. We trained two classes of pixels, namely CDR pixels and Background pixels, where the latter also included non-CDR actin staining, such as stress fibres. We saved the resulting classifier file to disk and used it in CellProfiler's ClassifyPixels module (see below). The image analysis was performed using CellProfiler. We describe the main steps of the CellProfiler pipeline; for more details please download the full pipeline, which runs with version 2.0.11710 of CellProfiler (http://cellprofiler.org/previous_releases/). We segmented nuclei in the Dapi image by means of an automated threshold (IdentifyPrimaryObjects module with the RobustBackgroundGlobal thresholding method). We used the MeasureImageQuality module to measure the focus quality of the Dapi image (ImageQuality_PowerLogLogSlope_nucleus); this value was used in our downstream analysis in R (see below) to reject out-of-focus images. Using the segmented nuclei as seeds, we applied a watershed algorithm on a smoothed and background corrected CellMask image in order to segment the cells (IdentifySecondaryObjects module with the Watershed-Image method). We shrank the cell objects by 5 pixels in order to avoid cell boundaries, because they were often misclassified as CDRs in the Phalloidin image. Using the FilterObjects module we rejected cells that were of very small size (less than 2500 pixels), had too dim Phalloidin staining (less than 200 mean value; the maximum being 4095 for our 12 bit data) and had too many Phalloidin pixels (more than 500) that were saturated (grey value = 4095). Our starting point for the segmentation of CDRs was a probability image generated by CellProfiler's ClassifyPixels module. The Classify-Pixels module applies the ilastik classifier (see above) on the Phalloidin image and outputs a new image with intensity values between zero and one. The closer the value is to one, the more certain the classifier is that the respective pixels belongs to a CDR. We applied a threshold of 0.51 to the probability image in order to obtain a binary image. (Note: As long as the binary image represents CDR pixels, the exact threshold value is not critical. For example, lowering the threshold slightly would slightly increase the number of CDR pixels in all images; as our final score of the screen is computed relative to the negative control siRNA (siEGFP) this increase would cancel out.) To distinguish CDRs from noise we performed a morphological

closing operation with a diameter of 7 pixels to connect neighbouring CDR pixels and then rejected all connected components (CDR objects) with an area of less than 25 pixels. For all cells—that were kept according to our filtering criteria described above—we computed the number of CDR-positive pixels and then computed the average value of all cells in the image as the final readout ("CDR_PerImage_RawScore").

The final output of our CellProfiler analysis was one table (Supplementary Data 2) where each row contained information about one image set. This "per-image" table was further analysed using R. First, we performed a per-image quality control by rejecting out-of-focus images (ImageQuality_PowerLogLogSlope_nucleus <−2) and by rejecting images with very few cells (Count_Cells <20). Then we performed a per-plate batch correction

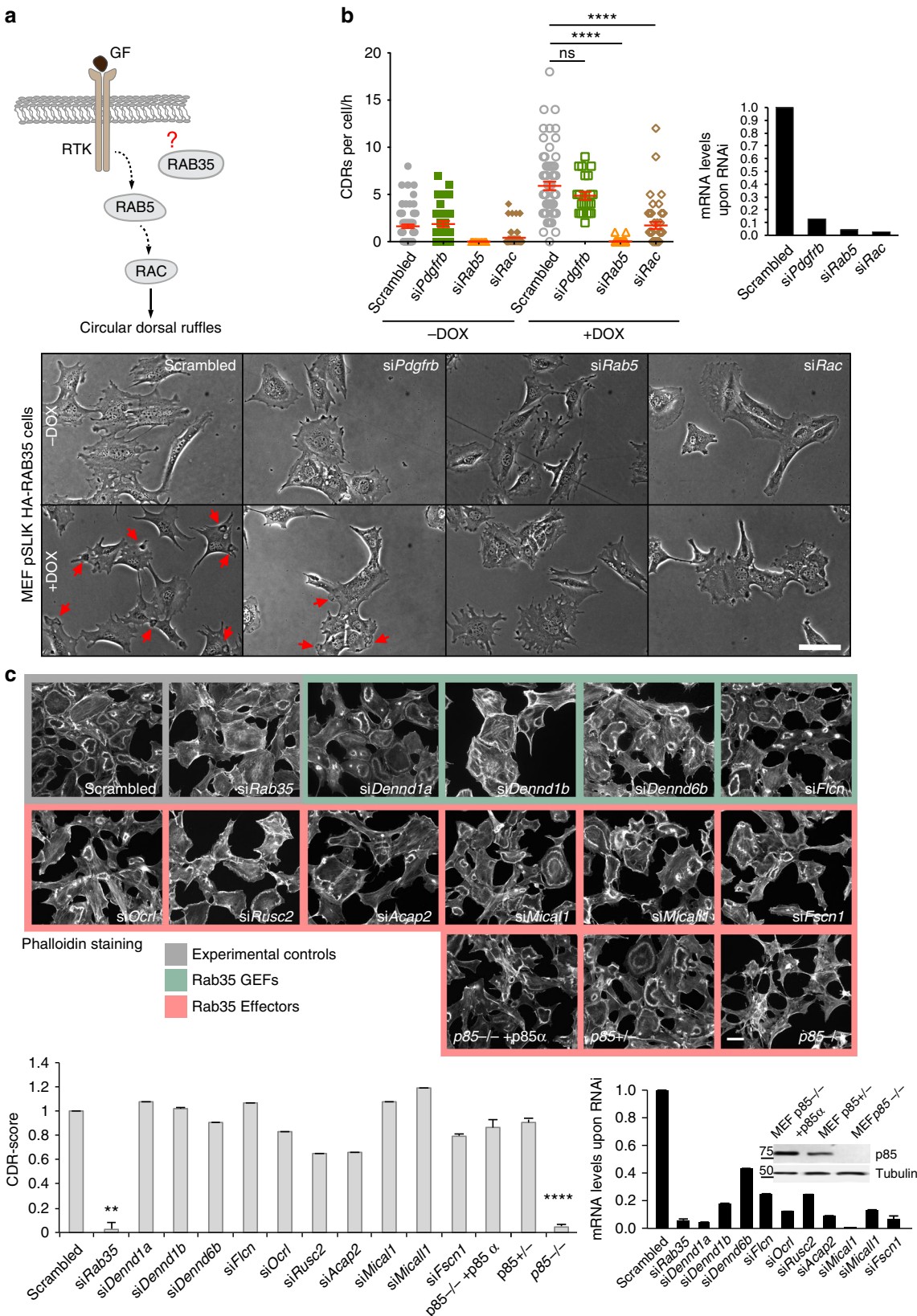

**Fig. 7** Molecules acting in concert with RAB35 to promote Dorsal Ruffles formation. **a** Schematic representation of the signalling pathway leading to CDR formation. **b** RAB5 and RAC are required for RAB35-dependent CDR formation but PDGFRB is dispensable. pSLIK-HA-RAB35-human-MEFs were silenced for *Pdgfrb*, or *Rab5* or *Rac1* and treated or not with doxycycline to induce the transgene expression. CDR formation was monitored by time-lapse phase contrast microscopy. Still phase contrast images from time-lapse sequence (Supplementary Movie 13) are shown. Red arrows indicate CDRs. Scale bar, 100 μm. The number of CDRs/cell/h was reported as mean ± SEM (>30 cells/condition, three independent experiments). ****$p < 0.0001$; ns not significant, paired Student's *t*-test. The silencing of genes was verified by qRTPCR. **c** Silencing of Rab35 GEFs or effectors. MEF cells were interfered for the indicated Rab35 GEFs or effectors and tested for their ability in forming CDRs upon PDGF stimulation. Representative Phalloidin-stained images are reported for each experimental condition. Scale bar, 50 μm. CDRs were manually counted and normalized against the scrambled sample. Data are the mean ± SD (n > 200 cells/condition in three independent experiments). ****$p < 0.0001$; **$p < 0.01$, paired Student's *t*-test. The silencing of the targeted genes was verified by qRTPCR. In the case of p85, double p85α and β KO MEFs[47] were employed and the loss of p85 isoforms was confirmed by western blotting. In the latter case, as control we used both p85α+/−p85β−/− MEFs (p85−/+, derived from sibling mice) as well as p85α−/−p85β−/− MEF in which p85α was re-expressed

of the CDR_PerImage_RawScore (see above). For each plate, we divided the CDR_PerImage_RawScore of all images by the average CDR_PerImage_RawScore of all images that were treated with our negative control siRNA (siEGFP), yielding the CDR_PerImage_RatioScore. Finally, for each siRNA we computed the mean value of the CDR_PerImage_RatioScore for all images where the respective treatment was applied, yielding the final CDR-Score, which is given in the table (Supplementary Data 2).

**RNA interference experiments in six-wells format.** Cells were reverse transfected by mixing 10 nM of specific siRNAs with 500 μl of Optimem and 5 μl of Lipofectamine RNAiMAX Transfection Reagent. For each experiment, scrambled siRNAs were used at the same concentration as specific siRNAs. Silencing efficiency was controlled by qRTPCR.

**Quantitative RTPCR analysis.** Total RNA was extracted using RNeasy Mini kit (Qiagen). SuperScript VILO cDNA Synthesis kit from Invitrogen was used for reverse transcribe 1 μg of total RNA.

Gene expression was analysed as in ref. [65]. Sequence-specific primers and probes were selected from the Assay-on-Demand products (TaqMan® gene expression assay; Applied Biosystems). Primer assay IDs were:

GAPDH, Hs99999905_m1 and Mm99999915_g1; RAB35, Hs00199284_m1 and Mm01204416_m1; PDGFRB, Mm00435546_m1; RAB8a, Mm00445684_m1; RAB8b, Mm00557812_m1; RAB5a, mm01278246_m1; RAB5b, Mm00834147_g1; RAB5c, mm00659190_m1; RAC1, mm01201653_mh; DENND1a, Mm00620186_m1; DENND1b, Mm01285090_m1; DENND1c, Mm01227405_m1; DENND6b, Mm01308059_m1; FLCN, Mm00840973_m1; OCRL, Mm00623482_m1; RUSC2, Mm01349595_m1; ACAP2, Mm01342334_m1; MICAL1, Mm00506780_m1; MICALL1, Mm01300206_m1; FSCN, mm00456046_m1.

**Western blotting.** Cells were lysed in RIPA buffer containing proteases and phosphatases inhibitors. Total cell lysates were resolved by gel SDS-PAGE, transferred onto Protran Nitrocellulose Transfer membrane (Whatman), probed with the appropriate antibodies and visualized with ECL Plus western blotting detection reagents (GE Healthcare). Membranes were blocked for 1 h in TBS/0.1% Tween/5% BSA for antibodies recognizing phosphorylated proteins or in TBS/0.1% Tween/5% milk for antibodies recognizing the total proteins. Primary antibodies incubation was performed according to the manufacturer's instructions. Secondary antibodies were diluted 1:10,000 and incubated for 1 h at RT.

Uncropped gels of all immunoblots are included in Supplementary Figures 5–8.

**Immunofluorescence.** Cell fixation was performed in 4% PFA and permeabilization in 0.1% Triton X-100 and 1% BSA for 10 min. After 1× PBS wash, samples were incubated with primary antibodies for 1 h at room temperature. Coverslips were washed in 1× PBS before the secondary antibody incubation for 1 h at room temperature, protected from light. FITC-Phalloidin was added in the secondary antibody step, where applicable. After 1× PBS washing step, nuclei were stained with 0.5 ng/ml Dapi and the plasma membrane was marked with CellMask Deep Red staining where indicated. Samples were mounted on glass slides in anti-fade mounting medium (Mowiol). Antibodies were diluted in 1× PBS and 1% BSA. Images were acquired by a wide-field fluorescence microscope or a confocal microscope.

**CDR formation assay.** To evaluate the formation of CDRs, MEF cells were serum starved for 2 h, stimulated with PDGF (20 ng/ml) for 10 min and fixed in 4% PFA for 10 min. After FITC-Phalloidin staining, the number of CDR-positive cells was counted and normalized to an internal control in order to generate the CDR score. The experimental procedure in HeLa cells was slightly different: cells were seeded on 0.5% Matrigel-coated coverslips, serum starved overnight and stimulated with HGF (100 ng/ml) for 10 min. At least 200 cells per experimental condition were analysed. Data are the mean of at least three independent experiments.

**Live monitoring of CDRs and analysis of cell protrusions.** Cells were seeded in six-wells plate ($2 \times 10^4$ cells/well) and cultivated in complete medium for 48 h. Where indicated, doxycycline 2.5 mg/ml was added 16 h before the experiment in order to induce RAB35 expression. Medium was refreshed before starting the time-lapse microscopy session. An Olympus ScanR inverted microscope with ×20 objective was used to take phase contrast and fluorescence images every 30″ over a 1 h period, otherwise differently indicated. The assay was performed using an environmental microscope incubator set to 37 °C and 5% $CO_2$ perfusion. Doxycycline was maintained in the media for the total duration of the time-lapse experiment. The number of CDRs per cell, formed during the live imaging session, was counted and reported in dot plot graphs. Each assay was done three times and at least 25 cells/condition were counted in each experiment. Where indicated, LY294002 (20 μM) was added 2 h before imaging. To capture the dynamics of protrusion formation in PDGF-treated or RAB35-ectopically expressing cells, movies generated by time-lapse microscopy were analysed by the Multiple Kymograph plugin of ImageJ software.

**Dextran internalization assay.** Cells were plated on un-coated glass coverslips and cultured in complete medium for 24 h. MEF shCtrl and shRAB35 cells were serum starved for 2 h and incubated concomitantly with 1 mg/ml TMR-dextran (70,000 MW) and PDGF (20 ng/ml) for 1 h at 37 °C. After four washing steps in ice-cold PBS, samples were fixed in PFA 4% and stained with Dapi. In the case of MEF pSLIK-HA-RAB35 cells, doxycycline was added to the media 16 h before performing the experiment and TMR-dextran (70,000 MW) was incubated for 1 h at 37 °C before cell fixation. Images were acquired by a wide-field fluorescence microscope. The dextran uptake was quantified by measuring the mean value of the corrected total cell fluorescence per cell in all the different experimental conditions. CTCF = integrated density – (area of selected cell × mean fluorescence of background readings). Three biological replicates were performed and at least 40 cells/condition were counted in each experiment.

**Boyden chamber and Matrigel invasion assay.** Cells were seeded into the upper compartment of a transwell or a Matrigel invasion chamber with 8 μm pore size filters (Corning) and the chemoattractant (PDGF 20 ng/ml, HGF 100 ng/ml, FBS 10% or EGF 100 ng/ml) was added to the lower chamber. After 20 h of incubation at 37 °C, samples were fixed in 4% PFA and cells migrated to the lower side of the microporous membrane were Crystal violet stained. Cells on the top of the membrane were scraped away. The average number of migrating cells per each experimental condition was counted and reported.

**Random migration assay.** The capability of cells in moving in the absence of any chemotactic stimulus was monitored by time-lapse phase contrast microscopy. Cells were seeded sparsely in a six-wells plate ($1 \times 10^4$ cells/well) in complete media and where indicated, supplemented with doxycycline (2.5 μg/ml) 16 h before starting the experiment. Random cell motility was monitored over a 24 h period. Pictures were acquired every 5 min from 6 positions/condition, using a motorized Olympus Scan^R inverted microscope with ×4 objective. All the experiments were performed using an environmental microscope incubator set to 37 °C and 5% $CO_2$ perfusion. Single cells were manually tracked using Manual Tracking Tool ImageJ software plugin. Migration plot and relative parameters were obtained by Chemotaxis Tool from Ibidi.

**Chemotaxis assay.** The ability of cells in responding to a chemotactic stimulus was tested by time-lapse phase contrast microscopy using fibronectin-coated 2D chemotaxis μ-slides from Ibidi. $2 \times 10^3$ cells were loaded into the central transversal channel and incubated at 37 °C for 30 min in order to allow the adhesion to the substrate. A PDGF-BB gradient (0–20 ng/ml) was generated according to the manufacturer's instructions. Cell motion was recorded using a motorized Olympus Scan^R inverted microscope with ×10 objective for a 24 h period, taking pictures every 5 min. All the assays were performed using an environmental microscope incubator set to 37 °C and 5% $CO_2$ perfusion. Chemotactic tracks were obtained using the Manual tracking ImageJ software plugin

and the chemotaxis plots and migratory parameters were obtained with the Chemotaxis Tool from Ibidi.

**Cell migration through an array of pillars**. The array of pillars was fabricated with two-photon polymerization on glass substrates. The structure consisted of two components: a dense micropillar array and wall-like barriers. The structure

components were first designed using CAD software (Autodesk Inventor, Autodesk). The inter-pillar distance was set to 4 μm in order to ensure the mechanical constriction of the cells during their migration through the array and the pillar thickness was set to 3 μm in order to provide the necessary mechanical stability to the structure. The barriers aimed to prevent cells from migrating around the pillar array (instead of migrating through it) or cells that would leave the array by

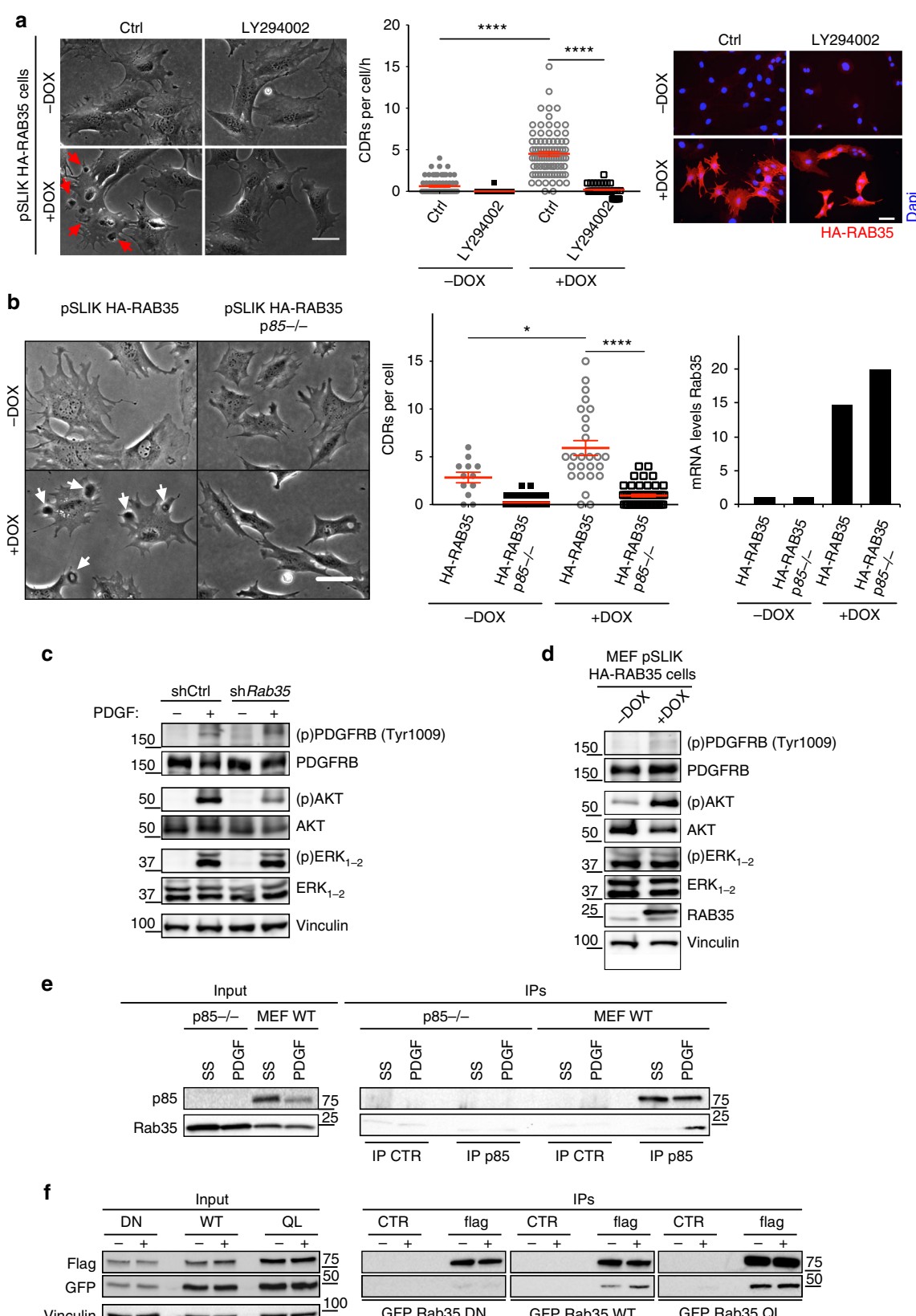

**Fig. 8** A RAB35/PI3K axis is necessary for triggering CDR formation. **a** PI3K inhibition impairs RAB35-induced CDR formation. pSLIK-HA-RAB35-human-MEFs were treated or not with doxycycline to induce the transgene expression and incubated with vehicle or LY294002. Still, phase contrast images from time-lapse sequence (Supplementary Movie 14) are shown. Red arrows indicate CDRs. Scale bar, 50 μm. The number of CDR/cell is the mean ± SEM (n > 30 cells/condition, three independent experiments). ****p < 0.0001. RAB35 expression was verified by IF. **b** Genetic ablation of the regulatory subunits of PI3K abrogates the RAB35-dependent CDR formation. Control or p85α−/− pβ−/− double KO MEFs (p85−/−)[47] were infected with pSLIK-HA-RAB35 to express RAB35 in a doxycycline-inducible fashion imaged by time-lapse phase contrast microscopy. Left: Still phase contrast images from time-lapse sequence (Supplementary Movie 15). Red arrows indicate CDRs. Scale bar, 50 μm. Middle graph: The number of CDR/cell is the mean ± SEM (n = 50 cells/condition, two independent experiments). ****p < 0.0001; *p < 0.05. Right graph: RAB35 expression was verified by qRTPCR. Data are the fold increase of RAB35 mRNA levels of doxycycline-treated with respect to untreated ones. **c, d** RAB35 regulates AKT activity. **c** Serum-starved control (shCtrl) and *Rab35*-silenced (shRab35) cells were stimulated or not for 10′ with PDGF. Cell lysates were analysed by immunoblotting to detect the indicated total or phosphorylated proteins. Vinculin is a loading control. **d** Lysates from pSLIK-HA-RAB35-human-MEFs treated or not with doxycycline were immunoblotted as in **c. e** RAB35 co-immunoprecipitates with endogenous p85α. Serum-starved p85α−/− pβ−/− double KO cells (p85−/−) and control MEFs were stimulated or not for 10′ with PDGF. Cell lysates (1 mg) were immunoprecipitated with anti-p85α monoclonal antibody or irrelevant immunoglobulin G (IgG). Cell lysates (50 μg) and immunoprecipitates (IPs) were immunoblotted with the indicated antibodies. **f** Active RAB35 interacts with p85α. Serum-starved HeLa cells transfected with RAB35 WT, or RAB35 S22N or RAB35Q67L fused to EGFP were mock treated (−) or stimulated with HGF (+) for 10′. Cell lysates (1 mg) were immunoprecipitated with anti-p85α antibody or irrelevant immunoglobulin G (IgG). Lysates (50 μg) and immunoprecipitates (IPs) were immunoblotted with the indicated antibodies. *p*-values are from paired Student's *t*-test

moving in transverse direction. The structural elements of the pillars and barriers (mesh slicing and hatching) and printing parameters of the whole structure (laser power, scan speed) were designed with the programme DeScribe v.2.4.4 (Nanoscribe GmbH), which creates a file readable by the laser lithography system. The sample fabrication was completed with a 3D laser lithography system (Photonic Professional GT, Nanoscribe GmbH, Germany) used in conventional oil-immersion mode. This involved passing a two-photon laser (780 nm laser, laser power 12.5 mW, scan speed 800 μm/s) from a ×63 objective lens through a layer of oil beneath the glass substrate, polymerizing the photoresist on the top side of the glass substrate. The photoresist chosen was a biocompatible[66], organic–inorganic hybrid polymer (Ormocomp®, MicroResist Technology GmbH). The resist was drop-casted on the glass substrates and pre-baked at 80 °C for 2 min. After laser writing, the resist was baked at 130 °C for 10 min, developed in OrmoDev® developer (MicroResist Technology GmbH) for 10 min to remove the un-polymerized resist and rinsed in isopropanol (IPA) followed by drying in a critical point dryer (Automegasamdri R 915B, Tousimis).

The device used to create a chemotactic gradient was a commercially available microfluidic chamber (sticky-Slide Chemotaxis, Ibidi) which enables the creation of a stable, linear gradient for more than 24 h. The glass substrate with the microstructures printed on top was attached on the self-adhesive underside of the chamber, with the micropillar array placed in the middle area. The chamber was filled with 70% ethanol for 1 h for sterilization, washed three times with 1× PBS for 5 min and incubated with fibronectin (10 μg/ml) for 1 h at 37 °C. Afterwards, fibronectin was gradually replaced by 1× PBS and finally by cell medium. $10^4$ cells were seeded from the inlets of the cell pool area (Fig. 4d) and were gently distributed over the cell pool area, with attention that they do not float over the pillars. The chambers were incubated 2 h at 37 °C and once the cells were spread, the chemoattractant (PDGF-BB) was added on the cell exit area at a concentration of 20 ng/ml. Cell motion was recorded using a motorized Olympus Scan^R inverted microscope with 10 × objective for a 24 h period, taking pictures every 10 min. All the assays were performed using an environmental microscope incubator set to 37 °C and 5% CO₂ perfusion.

**1D migration on micropatterns**. Micropatterns of fibronectin-coated lines (10 μm of width) were fabricated using photolithography as previously described[67]. Briefly, the glass surface was first activated with plasma (Plasma Cleaner, Harrick Plasma) and then coated with cell repellent PLL-g-PEG (Surface Solutions GmbH, 0.5 mg/ml in 10 mM HEPES pH 7.3). After washing with PBS and deionized water, the surface was illuminated with deep UV light (UVO Cleaner, Jelight) through a chromium photomask (JD-Photodata). Finally, coverslips were incubated with fibronectin (Sigma, 25 μg/ml in 100 mM NaHCO₃ pH 8.4). Cells were detached using EDTA 0.02% (Versane; GIbco) and left to attach on micropatterns. Afterwards, the coverslips were mounted in 35 mm magnetic chambers (Chamlide) for imaging. Cell locomotion was monitored by using a motorized Olympus Scan^R inverted microscope with ×10 objective for a 10 h period, acquiring images every 5 min. All the experiments were performed using an environmental microscope incubator set to 37 °C and 5% CO₂ perfusion.

**Preparation of fibronectin-coated suspended fibres**. Arrays of fibronectin-coated suspended fibres were prepared as previously described[68].

**Collagen invasion assay**. Type I collagen, at the final concentration of 2.3 mg/ml, was diluted in DMEM, Hepes (50 mM), NaOH (5 mM) and NaHCO₃ (0.12%) in the presence of HGF and EGF (100 ng/ml) and polymerized in homemade migration chambers (0.4-mm-thick) at 37 °C for 1 h. Cells were seeded on the top of the collagen matrix and the chambers were sealed with dental glue and incubated

at 37 °C, 5% CO₂. Cell locomotion through the 3D geometrically defined environment was followed by time-lapse phase contrast microscopy using a Nikon Eclipse TE2000-E microscope equipped with an incubation chamber for temperature and CO₂ control. Images were acquired every 10 min for a 48 h period. Cell trajectories were manually tracked using Manual Tracking Tool ImageJ software plugin. Invasive parameters were extrapolated by Chemotaxis Tool from Ibidi.

**MCF10.DCIS.com cells 3D invasion assay**. Acini were grown in 3D basement membrane cultures according to the standard protocol[69]. Briefly, Nunc Lab-Tek II 8-wells chamber slides were coated with a mix of collagen/Matrigel and allowed to solidify for 1 h at RT. Single cells were seeded in growing media supplemented with 2% of Matrigel and 5 ng/ml of EGF ($2.5 \times 10^3$ cells/well). Four days after seeding, the formation of invasive outgrowth was triggered by adding HGF (100 ng/ml) to the media. Phase contrast images were collected before and after growth factor stimulation. The percentage of 3D acini having invasive outgrowth was calculated. At least 60 spheroids per experimental condition were analysed.

**Flow cytometric analysis**. Cell surface levels of PDGF Receptor B were analysed by flow cytometry as follow. Briefly, $5 \times 10^5$ cells were blocked in 1× PBS + 1% BSA for 30 min at 4 °C, incubated with the anti-Mouse CD140b (PDGF Receptor b) PE for 30 min at 4 °C and fixed with 4% PFA for 15 min on ice. Data were acquired with the FACSCanto (Becton Dickinson) flow cytometer and analysed with FlowJo version 4.6.2 (Treestar).

**Automated immunohistochemistry with Leica BOND-MAX**. Immunohistochemical staining was performed as follows: 3-μm-thick sections were prepared from formalin-fixed paraffin-embedded TMA tissue blocks and dried in a 37 °C oven overnight. The sections were placed in a Bond Max Automated Immunohistochemistry Vision Biosystem (Leica Microsystems GmbH, Wetzlar, Germany) according to the following protocol. First, tissues were deparaffinized and pre-treated with the Epitope Retrieval Solution 2 (pH9) at 100 °C for 20 min. After washing steps, peroxidase blocking was carried out for 10 min using the Bond Polymer Refine Detection Kit DC9800 (Leica Microsystems GmbH). Tissues were again washed and then incubated for 30 min with the primary antibody diluted (1:50 anti-Rab35, Abcam ab 152138) in Bond Primary Antibody Diluent (AR9352). Subsequently, tissues were incubated with post primary and polymer for a total of 16 min and developed with DAB-Chromogen for 10 min and counterstained with haematoxilin for 5 min.

Human samples were collected with inform consent between 1995 and 2010, unfortunately, without recording their clinico-pathological features, thus impeding the analysis of RAB35 expression levels with relevant tumour properties.

**Statistical analysis**. Student's unpaired *t*-test was used for determining the statistical significance. Significance was defined as *p < 0.05; **p < 0.01; ***p < 0.001 and ****p < 0.0001. Statistic calculations were performed with GraphPad Prism Software. Data are expressed as mean ± SEM, unless otherwise indicated.

**Data Availability**. The authors declare that all data supporting the findings of this study are available within the article and its Supplementary Information files or from the corresponding author upon reasonable request.

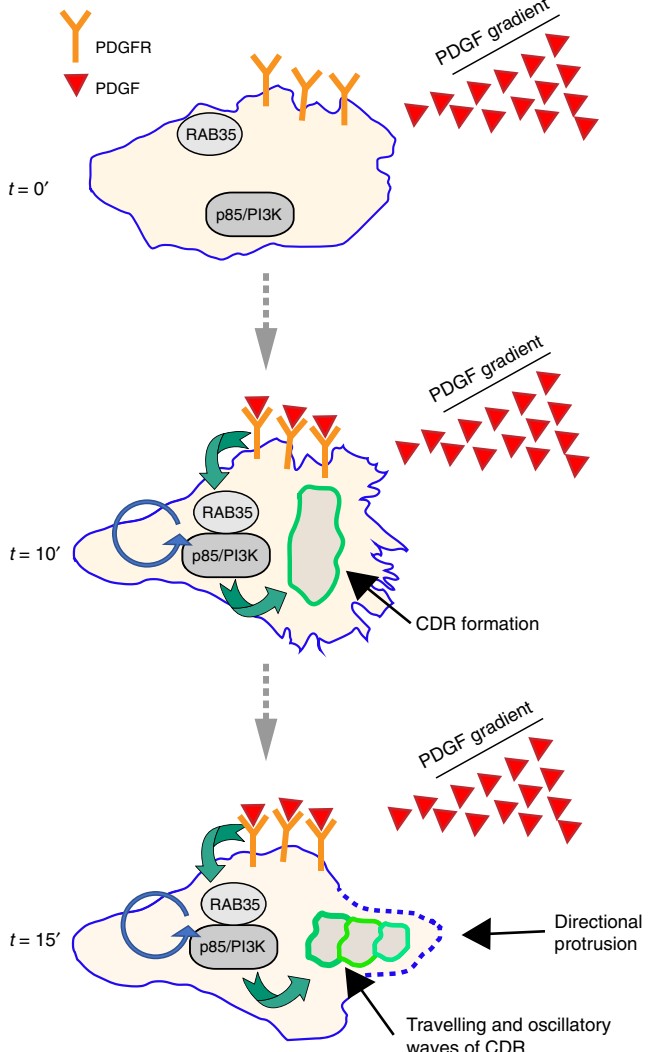

**Fig. 9** Model depicting the role of RAB35/p85–PI3K in CDR formation. Model depicting the role exerted by the RAB35–p85/PI3K axis in promoting the formation of excitable CDRs required to steer cells in response to PDGF (or HGF) stimulation. Top panel depict a mouse embryo fibroblast not exposed to PDGF gradients. RAB35 and p85/PI3K are not interacting. Middle panels, after few minutes upon exposure to PDGF (or HGF not depicted) gradients, PDGFR activation ensue, leading to the rapid formation of a RAB35–p85/PI3K complex. This cause the activation of PI3K that act upstream of RAC1 (not depicted) leading to the formation of apically localized CDRs. Lower panels, CDRs kinematics display a behaviour of oscillating and recurrent wave that travels along or expand centrifugally. Their formation is temporally correlated with the extension of directed cell protrusions. Thus, CDR display all the feature of an excitable oscillatory system that can be biased following chemotactic cues and contribute to stir cells along a chemotactic gradient

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

## Acknowledgements

We thank Giovanna Iodice for technical assistance with the IHC, and Arnaud Echard for critically reading the manuscript and providing key RAB35 reagents. This work has been supported by: the Associazione Italiana per la Ricerca sul Cancro (AIRC-IG#18621) to G. S., (AIRC-IG#18988), to P.P.D.F., (AIRC-IG#20716) to N.G. and (AIRC-IG-11904) to S. P.; the International Association for Cancer Research (AICR-14-0335); the European Research Council (Advanced-ERC#268836); Ministry of Health (RF-2013-02358446) to G.S.; MCO 10.000, and the Italian Ministry of University and Scientific Research (MIUR) to P.P.D.F. and S.P. S.C. and C.M. were supported by AIRC and Fondazione Umberto Veronesi fellowships, respectively.

## Author contribution

C.S designed, performed and analysed data and wrote the paper; C.M. performed and analysed data; B.N. and C.T. designed and developed software analysis tools for the imaging-based screening; A.P. and E.F. performed and developed assays of 3D cell migration; G.M.E. performed chemotactic assays; P.N. and P.M. performed and analysed micro-printed-based migratory assays; A.D. performed cellular biochemical experiments: C.G. and N.G. performed and analysed suspended fibres assays; C.L., S.P., G.B. and P.P. D.F. performed and interpreted the cancer data and IHC; A.F. and M.P. designed and build the forest of pillars. G.S. designed, analysed data and wrote the manuscript.

## Additional information

**Competing interests:** The authors declare no competing interests.

