## [Peer Review File · Nature Communications]

Reviewers' comments:

Reviewer #1 (Remarks to the Author):

This manuscript describes the role of Rab35 in the formation of circular dorsal ruffles (CDRs) and their involvement in migration. Overall, the manuscript is well written and the experiments are well done, presented and controlled. The involvement of Rab35 in the formation of CDRs is clearly documented and makes no doubt. The authors also clearly show that Rab35 acts through its effector p85 to form CDRs and suggest that their downstream target in this pathway is the PI3K and Akt pathway.

While this manuscript gives new information regarding a poorly understudied structure involved in cell migration and is, for this reason interesting and important it would need major revisions. Indeed, the author failed to convince me that 1) the molecular mechanism is indeed going through Pi3K and Akt, and that 2) the effect of Rab35 on the various migration assays tested is induced by a specific effect on CDR formation.

For the Akt/PI3K part, to demonstrate beyond any doubts that Rab35 acts uniquely through the Akt pathway, the author should perform epistatic-like experiments, if possible. For example, showing that a constitutive active Akt can bypass the requirement of Rab35. In addition, the use of the LY294002 compound for the PI3K/Akt characterization is problematic. Indeed, this drug is not entirely specific and the authors should confirm their observation with another compound targeting the pathway, ideally an inhibitor of Akt, or with Akt knock-down (kd). Furthermore, the authors claim that the effect of Rab35 is independent of the function of Rab35 in endocytosis, but the data are not really conclusive. Rab35 kd, leads to default of micropinocytosis, which might be linked to the more traditional trafficking function of Rab35. The author should at least show that blocking the Akt/PI3K affects CDRs without impacting on micropinocytosis.

The effect of Rab35 on the various mode of migration is quite well documented, but the authors did not clearly demonstrate that it is due to a default of formation of CDRs. This would be particularly interesting as the role of CDRs in 3D migration is poorly documented. To be more convincing, the authors should both 1) use a cell line that migrate in the same assays without forming CDRs and show that their migration is not affected by the depletion of Rab35 and 2) show that Rab35 depletion does not impact on the formation of other motile structures such as filopodia and lamellipodia.

Other minor points:

1) In Figure 4, the "pillar" migration assay panel is not very convincing. The effect and the phenotype are not obvious on the current image.

2) The experiment on the potential GEFs seems useless. If the author want to make a point about the function (or absence of function) of Rab35::GDP to GTP cycle, they should use dominant negative and constitutive active form of Rab35. It would be anyway interesting to know their effect on CDR formation and if they can rescue the loss of endogenous Rab35.

Reviewer #2 (Remarks to the Author):

In this manuscript, Salvatore et al demonstrate a novel role for Rab35 in the formation of circular dorsal ruffles and subsequent chemotactic cell migration. Through an RNAi screen, the authors determined which Rab GTPases were required for CDR formation. They found that knockdown of Rab35 inhibited growth factor-induced CDR formation, and overexpression of Rab35 induced CDR formation even in the absence of growth factor stimulation. Further, the authors determined that this is mediated by the Rab35 effector p85, which has a well-established role in formation of CDRs. In addition, the authors link Rab35-mediated CDR formation with directional cell migration. While beyond the scope of this manuscript, going forward it will be interesting to determine if Rab35 activation/GTP binding oscillates along with CDR formation. The manuscript is interesting and clearly written, and the figures are very clear. In all, this study advances the understanding of directional cell migration by

demonstrating a novel role for Rab35.

Specific Comments:

The authors suggest that the GTP/GDP cycling of Rab35 is likely involved in the oscillation of CDR formation. The experiments are performed with knockdown or overexpression of Rab35. Can the authors test mutated forms of Rab35 (constitutively active or dominant negative) in CDR formation and directional migration. This could potentially help address if it is the GTP state of Rab35 that is important, or if it is a GTPase-independent function.

In Figure 6C, what is the control for the p85^{-/-} MEFs? The other conditions are compared to the same MEF line transfected with a scrambled siRNA control. The p85^{-/-} MEFs are derived from a different embryo, which may not be directly comparable to the MEFs used elsewhere in the paper. A preferable control would be the p85^{-/-} lines + addback of p85, which should presumably rescue CDR formation. Also, can Rab35 overexpression in the p85^{-/-} cell lines induce CDR formation?

Perhaps Figure 1A (scheme for RNAi-based screen) could be moved to a Supplemental Figure.

The description of the screen (lines 90-117) is very helpful and clear, though could probably be moved to the Methods section. The description could be streamlined in the Results.

The "steering wheel" concept may be a bit overstated for the title. Rab35 does seem required for optimal directional migration, but it seems chemotactic migration is less efficient (rather than reversed/going in a different direction) in the absence of Rab35. Thus, it does not seem that "steering" is dramatically altered, but rather that there are impaired mechanics for targeting membrane dynamics in the direction of migration (via CDRs).

Perhaps a cartoon model could be included in Figure 7 that incorporates the receptor, Rab35, and p85/PI3K, and CDR formation.

Rab35 was described recently as a potentially oncogenic Rab protein. In the Conclusion, the authors could include a brief discussion of the implications of Rab35 directing CDR formation and directional migration in metastatic tumor cells. Do the tumor-associated mutations affect CDR formation?

Point-by-point Rebuttal

Reviewers' comments in black and italics, our reply in blue. The changes in the revised text are highlighted in yellow to facilitate reviewers' inspections.

REVIEWER 1

This manuscript describes the role of Rab35 in the formation of circular dorsal ruffles (CDRs) and their involvement in migration. Overall, the manuscript is well written and the experiments are well done, presented and controlled. The involvement of Rab35 in the formation of CDRs is clearly documented and makes no doubt. The authors also clearly show that Rab35 acts through its effector p85 to form CDRs and suggest that their downstream target in this pathway is the PI3K and Akt pathway.

R. We thank the reviewer for appreciating the novelty and solidity of our work.

While this manuscript gives new information regarding a poorly understudied structure involved in cell migration and is, for this reason interesting and important it would need major revisions. Indeed, the author failed to convince me that 1) the molecular mechanism is indeed going through Pi3K and Akt, and that 2) the effect of Rab35 on the various migration assays tested is induced by a specific effect on CDR formation.

1) For the Akt/PI3K part, to demonstrate beyond any doubts that Rab35 acts uniquely through the Akt pathway, the author should perform epistatic-like experiments, if possible. For example, showing that a constitutive active Akt can bypass the requirement of Rab35. In addition, the use of the LY294002 compound for the PI3K/Akt characterization is problematic. Indeed, this drug is not entirely specific and the authors should confirm their observation with another compound targeting the pathway, ideally an inhibitor of Akt, or with Akt knock-down (kd).

R. We apologize if we have not been sufficiently clear about the relationship among RAB35, p85/PI3K and AKT. We used AKT phosphorylation as a read out of PI3K activity. In the abstract, we erroneously stated (this statement has been corrected in the revised version of the abstract) that RAB35 impinges upon a p85/PI3K/AKT axis. However, our data indicates that RAB35 acts through p85/PI3K and is epistatic to this kinase, but not to AKT. In this respect, we showed that RAB35-induced CDRs are impaired following pharmacological inhibition of PI3K using LY294002, but also and more relevantly upon genetic removal of p85 isoforms. In the new Fig. 7B and Movie 15 of the revised manuscript, we showed that the expression of RAB35 in p85 null MEFs (see also replay to point #2 of Rev#2) fails in promoting CDRs. These results provide unequivocal genetic evidence in support of the epistatic relationship between RAB35 and p85/PI3K.

In addition, we have now performed new set of biochemical experiments that corroborate the genetic interaction study. Indeed, we showed that wild type RAB35 interacts with p85 only upon growth factor stimulation (new Fig. 7E of the revised manuscript). Furthermore, an inactive, dominant negative RAB35S22N mutant did not co-immunoprecipitated with p85 α . Whereas a constitutive active RAB35Q67L bound to p85 α even in under GF-starved conditions (new Fig. 7F of the revised manuscript). These results imply that stimulation with GF increases RAB35-GTP levels, leading to the interaction with p85 α /PI3K and subsequent activation of AKT. We included this finding in the revised Figure 7.

We used AKT phosphorylation as read out of p85/PI3K activity. Indeed, RAB35 removal impairs PDGF-induced AKT activation (Fig. 7C), consistent with its role in controlling the p85/PI3K pathway. Conversely, ectopic expression of RAB35 or of its oncogenic variants lead to increase AKT phosphorylation (Fig 7D and new Figure S4E of the revised manuscript). Prompted by reviewer's comments, we further investigate the role of AKT. To this end, we employed, as suggested, a pharmacological established strategy to inhibit AKT activity and test its impact on CDR formation. We used MK-2206, a specific allosteric AKT inhibitor¹. MK-2206 treatment had no effect on PDGF-induced CDR formation or on macropinocytosis (new Fig. S4C of the revised

manuscript), while GDC-0941, a more specific PI3K inhibitor abrogated these structures (new Fig. S4B of the revised manuscript). These findings are consistent with a recent set of results, showing that inhibition of AKT has no effect on macropinocytic-dependent dextran internalization². CDRs are indeed invariably leading to the formation of macropinocytic vesicles (Fig. 2B and 2D, see also Movie 1). This latter point is further discussed and clarified in the reply to point 2 below.

Thus, collectively our finding indicates that RAB35 acts upstream of and physically interact with p85/PI3K (see new Fig. S6, where a model summarizing our finding is depicted as requested by reviewer #2). The activation of the latter leads to increase phosphorylation of AKT, which however is not directly involved in CDR formation or macropinocytosis. Instead, we showed that RAB35/p85/PI3K axis requires RAC1, in keeping with the role of the latter kinase in regulating the activity of this GTPase and with the critical requirement of RAC1 for CDR formation (Fig. 6B) and macropinocytosis³⁻⁵.

2. Furthermore, the authors claim that the effect of Rab35 is independent of the function of Rab35 in endocytosis, but the data are not really conclusive. Rab35 kd, leads to default of micropinocytosis, which might be linked to the more traditional trafficking function of Rab35. The author should at least show that blocking the Akt/PI3K affects CDRs without impacting on micropinocytosis.

R. We thank the reviewer to allow us to clarify this issue. One well established function of RAB35 is its role in clathrin-dependent endocytosis and recycling⁶. By targeting, through siRNA, the set of RAB35 effectors that this GTPase uses in controlling these processes (see Fig. 6C), we showed that these molecules are not required for CDRs formation, arguing that the canonical clathrin-dependent and recycling activities of RAB35 are not involved in the regulation of these migratory protrusions. However, growth factor or RAB35-dependent CDRs invariably lead, after they enclosed, to the formation of macropinosomes in agreement with a bulk of literature indicating that these structures are site of macropinocytic internalization⁷⁻¹⁶. Thus, macropinocytic internalization is mechanistically and biochemical distinct from RAB35 role on clathrin-dependent internalization and recycling, but CDR and macropinocytosis are two inextricably linked aspects of the same phenomena. Consistently, inhibition of CDR by RAB35 silencing also leads to impair Dextran uptake (Fig. 2B), while CDR induction by RAB35 is invariably accompanied by increased Dextran uptake (Fig. 2D). Conversely, AKT inhibition does not affect CDR formation and has no impact on macropinocytosis (new Fig. S4C and ref. ²). We apologize if we have not been sufficiently clear on this point, and re-phrased our text to avoid possible misunderstanding (see newly revised abstract and text all changes are highlighted in yellow).

3) The effect of Rab35 on the various mode of migration is quite well documented, but the authors did not clearly demonstrate that it is due to a default of formation of CDRs. This would be particularly interesting as the role of CDRs in 3D migration is poorly documented. To be more convincing, the authors should both 1) use a cell line that migrate in the same assays without forming CDRs and show that their migration is not affected by the depletion of Rab35 and 2) show that Rab35 depletion does not impact on the formation of other motile structures such as filopodia and lamellipodia.

R. Agree. To clarify and reinforce the notion that RAB35 controls cell migration primarily in response to chemotactic cues (PDGF and HGF, in particular) that promote robust CDRs formation, we followed reviewer's suggestion, and analysed chemotaxis of mouse embryonic fibroblasts after RAB35 silencing in response to stimuli that poorly or do not induce CDR formation, such as serum or EGF, respectively. Firstly, we now show that addition of serum poorly induces CDR formation as compared to PDGF stimulation. Consistently, RAB35 silencing impairs only slightly chemotactic migration toward serum particularly when compared to the drastic inhibitory effect exerted in the presence of PDGF (new Fig. S2A and S2B of the revised manuscript). Even more remarkably, RAB35 loss has no impact on migration of MEFs toward EGF, which is proficient in

promoting chemotaxis, but unable to induce CDR formation (new Fig. S2C and S2D of the revised manuscript). Similarly, RAB35 was dispensable for directed migration in a scratch wound healing assays, i.e. under conditions in which cells move by kenotaxis in the absence of PDGF and CDRs (new Fig. S2E and Movie S9 of the revised manuscript). We included a new Fig. S2 in the revised manuscript and commented on this results on pg. 8-line 207-213 of the revised manuscript).

Finally, when we monitored cell protrusions dynamics either in the absence of PDGF, in cell moving randomly (Movie 2, compare control with siRAB35 cells, and Movie 8), or along chemotactic gradients of PDGF (Movie 7), or in wound healing assays (newly added Movie 9) we observed that RAB35-deficient MEFs readily extended lamellipodia and filopodia-like protrusions similarly to control cells, as predicted by the reviewer (see also Addendum Movie for Reviewers, where arrows point to the extension of lamellipodia which form indistinguishably and with a similar frequency in control and siRAB35 cells moving to close a wound. The Addendum Movie are a set of magnified frames extracted from the newly added Movie 9).

The sum of these findings strengthen the notion that RAB35 acts specifically in controlling CDR formation and CDR-dependent cell steering.

Other minor points:

1) In Figure 4, the “pillar” migration assay panel is not very convincing. The effect and the phenotype are not obvious on the current image.

R. Agree. The presence of the array makes it difficult to obtain high resolution images. We have provided examples of higher magnification of cells to show more clearly the extension of protrusions oriented along the chemotactic gradient with respect to those deviating from this angle. We have included these examples in the revised new Figure 4D of the manuscript.

2) The experiment on the potential GEFs seems useless. If the author want to make a point about the function (or absence of function) of Rab35::GDP to GTP cycle, they should use dominant negative and constitutive active form of Rab35. It would be anyway interesting to know their effect on CDR formation and if they can rescue the loss of endogenous Rab35.

R. The experiment with the siRNA targeting all known GEFs for RAB35 was performed in the attempt to identify upstream molecular determinants regulating RAB35. Indeed, as pointed out by the reviewer, the expression of RAB35 dominant negative prevents PDGF-induced CDRs (see new Fig. S4D of the revised manuscript, see also reply to point 1 of reviewer #2). Conversely, the fast cycling RAB35A151T and F161L mutants found in cancer¹⁷ promote constitutive CDRs formation (new Fig. S4E of the revised manuscript.)

REVIEWER 2

In this manuscript, Salvatore et al demonstrate a novel role for Rab35 in the formation of circular dorsal ruffles and subsequent chemotactic cell migration. Through an RNAi screen, the authors determined which Rab GTPases were required for CDR formation. They found that knockdown of Rab35 inhibited growth factor-induced CDR formation, and overexpression of Rab35 induced CDR formation even in the absence of growth factor stimulation. Further, the authors determined that this is mediated by the Rab35 effector p85, which has a well-established role in formation of CDRs. In addition, the authors link Rab35-mediated CDR formation with directional cell migration. While beyond the scope of this manuscript, going forward it will be interesting to determine if Rab35 activation/GTP binding oscillates along with CDR formation. The manuscript is interesting and clearly written, and the figures are very clear. In all, this study advances the understanding of directional cell migration by demonstrating a novel role for Rab35.

R. We thank the reviewer for the positive comment about our work.

Specific Comments:

1. *The authors suggest that the GTP/GDP cycling of Rab35 is likely involved in the oscillation of CDR formation. The experiments are performed with knockdown or overexpression of Rab35. Can the authors test mutated forms of Rab35 (constitutively active or dominant negative) in CDR formation and directional migration. This could potentially help address if it is the GTP state of Rab35 that is important, or if it is a GTPase-independent function.*

R. We tested a dominant negative RAB35S22N and two active RAB35A151T and F161L mutants found in cancer¹⁷. We found that RAB35S22N inhibits CDRs formation and PDGF-chemotaxis, while RAB35A151T and F161L mutants induces the constitutive formation of CDRs in the absence of growth factor stimulation (new Fig. S4D and S4E of the revised manuscript). This finding support the notion that GDP/GTP cycling is important for RAB35 to exert its function in CDR formation. We further showed that RAB35 interact with p85, which is essential for CDR formation and directional motility, in a GTP-dependent and growth factor-mediated fashion, reinforcing the notion that the GTP states is important for the function exerted by RAB35.

2. *In Figure 6C, what is the control for the p85^{-/-} MEFs? The other conditions are compared to the same MEF line transfected with a scrambled siRNA control. The p85^{-/-} MEFs are derived from a different embryo, which may not be directly comparable to the MEFs used elsewhere in the paper. A preferable control would be the p85^{-/-} lines + add back of p85, which should presumably rescue CDR formation. Also, can Rab35 overexpression in the p85^{-/-} cell lines induce CDR formation?*

R. Agree. In Figure 6C, we used as control both a different not matched preparation of wild type fibroblasts, as well as p85 α ^{-/+}p85 β ^{-/-} derived from sibling embryo with respect to the p85 α ^{-/-}p85 β ^{-/-} double KO cells. We also employed p85 α ^{-/-}p85 β ^{-/-} MEF in which p85 α expression was restored (see newly revised Fig. 6C and revised Methods describing the generation of mouse embryo fibroblasts). The p85 α ^{-/+}p85 β ^{-/-} MEFs form CDRs as efficiently as wild type cells. Genetically modified MEFs were kindly provided by Lew Cantley lab and have been characterized in term of actin dynamics and RAC activation in a previous work¹⁸.

RAB35 ectopic expression is no longer able to induce CDRs in p85^{-/-} cells (see new Fig. 7B and Movie 15 of the revised manuscript).

3. *Perhaps Figure 1A (scheme for RNAi-based screen) could be moved to a Supplemental Figure.*

R. Agree. We moved the RNAi-based flow chart to Supplementary information (Fig. S1A of the revised manuscript)

4. *The description of the screen (lines 90-117) is very helpful and clear, though could probably be moved to the Methods section. The description could be streamlined in the Results.*

R. We moved the indicated text to Methods.

5. The “steering wheel” concept may be a bit overstated for the title. Rab35 does seem required for optimal directional migration, but it seems chemotactic migration is less efficient (rather than reversed/going in a different direction) in the absence of Rab35. Thus, it does not seem that “steering” is dramatically altered, but rather that there are impaired mechanics for targeting membrane dynamics in the direction of migration (via CDRs). Perhaps a cartoon model could be included in Figure 7 that incorporates the receptor, Rab35, and p85/PI3K, and CDR formation.

R. We changed the title as follows: “A RAB35-p85/PI3K axis controls oscillatory apical protrusions required for efficient chemotactic migration”. We also included a cartoon in Fig. S6 of the revised manuscript as suggested.

6. Rab35 was described recently as a potentially oncogenic Rab protein. In the Conclusion, the authors could include a brief discussion of the implications of Rab35 directing CDR formation and directional migration in metastatic tumor cells. Do the tumor-associated mutations affect CDR formation?

R. We amended the discussion as per reviewers suggestions. As discussed in the reply to point 1 of this reviewer, we have now shown that the oncogenic RAB35 mutants also induce constitutive CDRs formation (new Fig. S4E of the revised manuscript).

References

1. Yap, T.A. *et al.* First-in-man clinical trial of the oral pan-AKT inhibitor MK-2206 in patients with advanced solid tumors. *J Clin Oncol* **29**, 4688-4695 (2011).
2. Palm, W., Araki, J., King, B., DeMatteo, R.G. & Thompson, C.B. Critical role for PI3-kinase in regulating the use of proteins as an amino acid source. *Proc Natl Acad Sci U S A* **114**, E8628-E8636 (2017).
3. Palm, W. *et al.* The Utilization of Extracellular Proteins as Nutrients Is Suppressed by mTORC1. *Cell* **162**, 259-270 (2015).
4. Koivusalo, M. *et al.* Amiloride inhibits macropinocytosis by lowering submembranous pH and preventing Rac1 and Cdc42 signaling. *J Cell Biol* **188**, 547-563 (2010).
5. West, M.A., Prescott, A.R., Eskelinen, E.L., Ridley, A.J. & Watts, C. Rac is required for constitutive macropinocytosis by dendritic cells but does not control its downregulation. *Curr Biol* **10**, 839-848 (2000).
6. Klinkert, K. & Echard, A. Rab35 GTPase: A Central Regulator of Phosphoinositides and F-actin in Endocytic Recycling and Beyond. *Traffic* **17**, 1063-1077 (2016).
7. Bernitt, E., Koh, C.G., Gov, N. & Dobereiner, H.G. Dynamics of actin waves on patterned substrates: a quantitative analysis of circular dorsal ruffles. *PLoS One* **10**, e0115857 (2015).
8. Sero, J.E., German, A.E., Mammoto, A. & Ingber, D.E. Paxillin controls directional cell motility in response to physical cues. *Cell Adh Migr* **6**, 502-508 (2012).
9. Hoon, J.L., Wong, W.K. & Koh, C.G. Functions and regulation of circular dorsal ruffles. *Mol Cell Biol* **32**, 4246-4257 (2012).
10. Zeng, Y., Lai, T., Koh, C.G., LeDuc, P.R. & Chiam, K.H. Investigating circular dorsal ruffles through varying substrate stiffness and mathematical modeling. *Biophys J* **101**, 2122-2130 (2011).
11. Peleg, B., Disanza, A., Scita, G. & Gov, N. Propagating cell-membrane waves driven by curved activators of actin polymerization. *PLoS One* **6**, e18635 (2011).
12. Gu, Z., Noss, E.H., Hsu, V.W. & Brenner, M.B. Integrins traffic rapidly via circular dorsal ruffles and macropinocytosis during stimulated cell migration. *J Cell Biol* **193**, 61-70 (2011).

13. Gao, X., Xing, D., Liu, L. & Tang, Y. H-Ras and PI3K are required for the formation of circular dorsal ruffles induced by low-power laser irradiation. *J Cell Physiol* **219**, 535-543 (2009).
14. Orth, J.D., Krueger, E.W., Weller, S.G. & McNiven, M.A. A novel endocytic mechanism of epidermal growth factor receptor sequestration and internalization. *Cancer Res* **66**, 3603-3610 (2006).
15. Frittoli, E. *et al.* The primate-specific protein TBC1D3 is required for optimal macropinocytosis in a novel ARF6-dependent pathway. *Mol Biol Cell* **19**, 1304-1316 (2008).
16. Lanzetti, L., Palamidessi, A., Areces, L., Scita, G. & Di Fiore, P.P. Rab5 is a signalling GTPase involved in actin remodelling by receptor tyrosine kinases. *Nature* **429**, 309-314 (2004).
17. Wheeler, D.B., Zoncu, R., Root, D.E., Sabatini, D.M. & Sawyers, C.L. Identification of an oncogenic RAB protein. *Science* **350**, 211-217 (2015).
18. Brachmann, S.M. *et al.* Role of phosphoinositide 3-kinase regulatory isoforms in development and actin rearrangement. *Mol Cell Biol* **25**, 2593-2606 (2005).

REVIEWERS' COMMENTS:

Reviewer #1 (Remarks to the Author):

The revised version adequately answers to my main concerns. In particular, Fig.S2 clearly demonstrates the specificity of the role of Rab35 and CDR in the migration process. The revised wording and the supplemental material concerning the role of the Akt pathway and about the potential interference of the role of Rab35 in trafficking and in migration is also properly addressed. As such, I support the publication of this revised version.

Reviewer #2 (Remarks to the Author):

The authors have addressed all of my comments .

Very minor points (should not preclude acceptance):

The authors should label the Input in the new Figure 7E.

The authors refer to Tables 1 and 2, but I could not find them in this submission? Are they included?

Minor proofreading is needed for English grammar.

POINT-BY-POINT REPLY TO REVIEWERS' COMMENTS:

Our comments are highlighted in yellow

Reviewer #1 (Remarks to the Author):

The revised version adequately answers to my main concerns. In particular, Fig.S2 clearly demonstrates the specificity of the role of Rab35 and CDR in the migration process. The revised wording and the supplemental material concerning the role of the Akt pathway and about the potential interference of the role of Rab35 in trafficking and in migration is also properly addressed.

As such, I support the publication of this revised version.

R. we thank the reviewer for the appreciation of our work and effort to address her/his concerns.

Reviewer #2 (Remarks to the Author):

The authors have addressed all of my comments

R. we thank the reviewer for the appreciation of our work and effort to address her/his concerns.

Very minor points (should not preclude acceptance):

The authors should label the Input in the new Figure 7E.

R. We change the figure 7E (that has become Figure 8E of the revised manuscript) accordingly.

The authors refer to Tables 1 and 2, but I could not find them in this submission? Are they included?

R. We included the Tables as Supplementary Data 1 and 2.

Minor proofreading is needed for English grammar.

R. We proof read the manuscript.